# SURVEYBENCH: HOW WELL CAN LLM(-AGENTS) WRITE ACADEMIC SURVEYS?

## ABSTRACT

Academic survey writing, which distills vast literature into a coherent and insightful narrative, remains a labor-intensive and intellectually demanding task. While recent approaches, such as general DeepResearch agents and survey-specialized methods, can generate surveys automatically (a.k.a. LLM4Survey), their outputs often fall short of human standards and there lacks a rigorous, reader-aligned benchmark for thoroughly revealing their deficiencies. To fill the gap, we propose a fine-grained, quiz-driven evaluation framework **SurveyBench**, featuring (1) typical survey topics source from recent 11,343 arXiv papers and corresponding 4,947 high-quality surveys; (2) a multifaceted metric hierarchy that assesses the outline quality (e.g., coverage breadth, logical coherence), content quality (e.g., synthesis granularity, clarity of insights), and non-textual richness; and (3) a dual-mode evaluation protocol that includes content-based and quiz-based answerability tests, explicitly aligned with readers' informational needs. Results show SurveyBench effectively challenges existing LLM4Survey approaches (e.g., on average 21% lower than human in content-based evaluation).

## 1 INTRODUCTION

Academic surveys are essential for both newcomers and experts to gain an authoritative understanding of fast-moving fields Zhang et al. (2025); Sapkota et al. (2025). Different from other long-form text generation tasks (e.g., wiki-style article generation Shao et al. (2024)), writing a high-quality academic survey is challenging. First, it needs to comprehensively cover a field's extensive and highly relevant literature (e.g., 5,200 publications of "Probabilistic methods" on arXiv). Second, it calls for meticulous and well-designed presentation, where (1) each chapter owns clear logical structures, (2) methods are precisely categorized, and (3) insights are deeply articulated (e.g., comparing the strengths and weaknesses). Besides, it needs to provide a forward-looking perspective, offering reasoned predictions of emerging trends and future directions. As shown in Figure 1, typical process often takes human writers months or even year to finish, which can produce high-quality surveys, but is (1) time-consuming, (2) costly, and (3) at risk of outdate due to the rapid scientific advance pace.

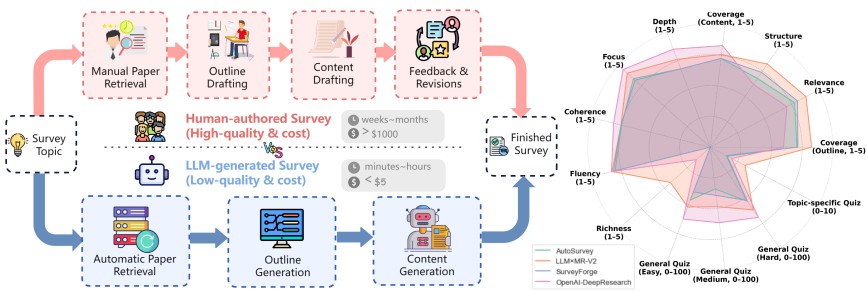

Figure 1: Human vs. Automatic LLM(-Agent) Based Survey Writing.

Recently, LLM-based agents have shown remarkable potential for automating academic survey writing. On one hand, general LLM agents with deep-research capabilities (e.g., OpenAI (2025), Google (2024), Du et al. (2025)) can retrieve, synthesize, and reason over large-scale relevant papers to draft comprehensive surveys with minimal human intervention (e.g., OPENAI-DEEPRESEARCH finishes a survey in minutes). On the other hand, LLM4Survey methods (e.g., AUTOSURVEY Wang et al. (2024), LLM×MAPREDUCE-V2 Wang et al. (2025), SURVEYX Liang et al. (2025)) explicitly

target the unique challenges of survey writing, which incorporate tailored mechanisms such as literature mining, automated citation management, and structured chapter planning. However, although these methods demonstrate promising scores in general metrics (e.g., ROUGE Wang et al. (2025), BERTScore Liang et al. (2025), citation density Yan et al. (2025)), compared to high-quality human-written surveys, their outputs still suffer from critical issues such as (1) imbalanced, outdated, or low-quality references, (2) incomplete or biased coverage of key techniques, (3) shallow insights, and (4) a lack of critical comparison or actionable takeaways.

Therefore, there is a pressing need for a rigorous, reader-aligned benchmark that can accurately reflect the survey-writing capability. Building such a benchmark poses several challenges: (1) a gap exists between many computer-science topics and the availability of representative high-quality surveys, which remain scarce and highly domain-specific, making it difficult to establish broad and fair reference standards; (2) survey content evaluation is inherently multi-faceted, requiring assessment of outline quality, content quality, and the richness of multimodal elements that aid understanding; and (3) existing LLM-as-judge evaluation struggles to capture the reader's perspective or to probe whether a survey genuinely informs (e.g., technical depth) and inspires (e.g., forward-looking insights).

In the real-world, *readers typically find surveys valuable when they provide clear answers to core research questions, such as technical solutions to specific problems or when they offer novel insights that inspire further exploration.* Inspired by this, we introduce **SurveyBench**, a fine-grained, quiz-driven evaluation framework with three main components: (1) Curated Benchmark Dataset: A collection of popular research topics paired with high-quality human-written surveys, covering a wide spectrum of computer science fields. (2) Dual-Setting Evaluation Protocol. Incorporating both human-reference-based evaluation (e.g., comparison against gold-standard surveys) and non-reference-based metrics (e.g., answerability via quiz-style evaluation). (3) Hierarchical Evaluation Dimensions: Capturing the full complexity of survey quality across outline structure (e.g., coverage completeness, logical organization) and content depth (e.g., synthesis granularity, insight articulation), richness (i.e., proportions of non-text elements like charts and diagrams).

To validate the effectiveness of SurveyBench, we evaluate OPENAI-DEEPRESEARCH alongside three survey-specific methods. Results show that while LLM-generated surveys demonstrate fluent and well-structured expression and basic instructional value, they still fall markedly short of human-written surveys in content metrics such as richness and in quiz-based assessments (especially topic-specific quizzes), underscoring the need for more targeted optimization in automatic survey writing.

Our main contributions are as follows:

- We introduce SurveyBench, a comprehensive benchmark for academic survey writing, covering representative topics drawn from 11,343 recent arXiv papers and 4,947 high-quality surveys.

- We propose a fully automated evaluation framework featuring (i) leakage-avoiding survey prompt design (e.g., fairness-guaranteed instructions), (ii) a fine-grained metric hierarchy for long-form survey evaluation, and (iii) quiz-driven validation to detect shallow or misleading content.

- We conduct an extensive empirical study benchmarking three survey-specific methods and OPENAI-DEEPRESEARCH, revealing substantial performance gaps in outline structure, content depth, and quiz-based answerability compared with human-expert written surveys.

## 2 AUTOMATIC SURVEY WRITING PIPELINES

As shown in Figure 2, LLM4Survey methods generally mimic the workflow of human authors.

**Publications Retrieval.** Most methods adopt embedding-based retrieval to gather relevant literature: *(i) Reference sourcing* collects candidate papers from offline or online databases. For instance, AUTOSURVEY (Wang et al., 2024) and SURVEYFORGE (Yan et al., 2025) use preprocessed embeddings of large-scale literature databases, while SURVEYX (Liang et al., 2025) combines its database with Google Scholar to capture recent works. OPENAI-DEEPRESEARCH (OpenAI, 2025) query open-access sources during reference generation. *(ii) Reference quality control* ensures relevance and coverage. SURVEYX expands keywords via semantic clustering and applies a two-stage embedding–LLM filtering process. SURVEYFORGE additionally considers time and impact by grouping papers by publication date and selecting top-cited works. *(iii) Reference preprocessing* structures the retrieved papers for downstream use. SURVEYX, for example, builds attribute tree templates (e.g., for reviews or methodology papers) and uses LLMs to extract and populate structured information.

Figure 2: Common Pipelines of Existing LLM4Survey Methods.

**Outline Generation.** Outline generation defines the logical structure of the survey: *(i) Initial generation* drafts outlines from retrieved references. To handle context limits, AUTOSURVEY and LLM×MAPREDUCE-V2 Wang et al. (2025) batch literature and merge multiple partial outlines. SURVEYX extends its attribute tree to guide second-level headings, while SURVEYFORGE leverages both topic-relevant papers and existing survey outlines. *(ii) Outline refinement* improves consistency and coverage. SURVEYX deduplicates and reorganizes headings, while LLM×MAPREDUCE-V2 applies entropy-driven convolution and best-of-N self-refinement for higher-quality outlines.

**Content Generation.** Content generation produces the final text: *(i) Initial generation* writes draft sections based on the outline and references. AUTOSURVEY and SURVEYFORGE generate subsections in parallel, SURVEYX uses a sequential approach to incorporate context from prior sections, and LLM×MAPREDUCE-V2 adopts a tree-based process that integrates leaf digests and subsection content. *(ii) Content refinement* enhances clarity, consistency, and citations. AUTOSURVEY and SURVEYFORGE refine sections with neighboring context, with AUTOSURVEY adding citation verification. SURVEYX retrieves from its attribute forest to filter and rewrite paragraphs.

## 3 SURVEYBENCH

In this section, we introduce the overall framework of SurveyBench (Figure 3), covering the benchmark construction process, key dataset statistics, and detailed evaluation procedure.

### 3.1 BENCHMARK CONSTRUCTION

Building an effective survey-writing benchmark faces two main challenges. First, the explosive growth of papers on arXiv and Google Scholar complicates topic selection, which must be (1) *typical*, covering mature areas with rich, influential literature; (2) *diverse*, spanning subfields such as reinforcement learning, multimodal learning; (3) *diagnostic*, exposing weaknesses like poor structure or shallow content. Second, evaluation demands fine-grained criteria beyond surface fluency*.

### 3.1.1 SURVEY TOPIC PREPARATION

We curate survey topics in three stages. First, we collect 127 candidates from authoritative computer science sources, including top conferences (e.g., ICLR, NeurIPS, CVPR, SIGMOD, SOSP), and refine them by removing duplicates and unifying terminology (e.g., merging "Brain-Computer Interfaces" and "Neural Coding"). Second, for each refined topic, we cluster recent arXiv papers from the past three months, computing embeddings (via the text-embedding-3-small model) from titles, abstracts, and key topics, and applying t-SNE for dimensionality reduction and visualization. As shown in Figure 4, we prune topics based on publication volume, conceptual diversity, academic influence (citations and top-tier venue presence), and semantic overlap. Finally, for each remaining topic, we sample 4947 survey papers from arXiv using "survey" or "review" keywords and further filter them by (1) *Impact* (citation counts from Semantic Scholar or arXiv-sanity) and (2) *Coverage Depth* (topical alignment with the retrieved papers), yielding an ultimate 20 well-vetted topics for benchmark usage.

---

*LLM-written surveys often excel in general NLP metrics but lack academic depth and rigor (Appendix B).

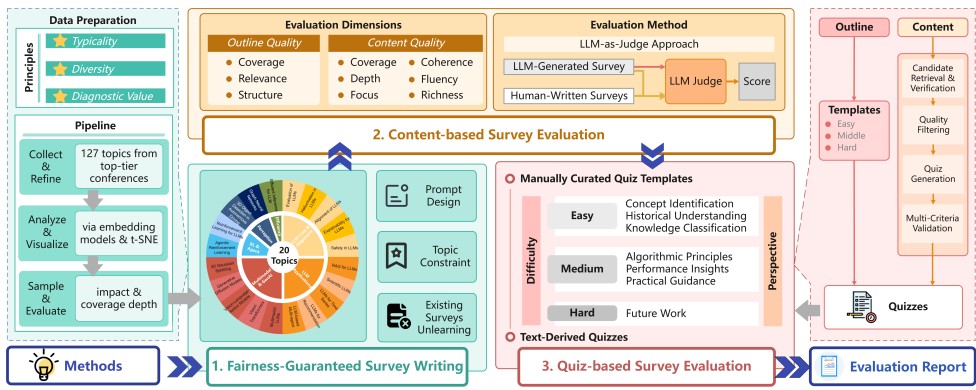

Figure 3: Overview of the Evaluation Framework.

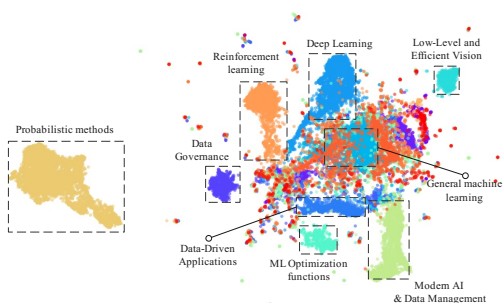

Figure 4: Example Publication Distribution. The dashed boxes showcase nine primary topics.

Table 1: Sampled Survey Statistics. This table shows the number of surveys sampled that belong to the nine primary topics shown in the left figure, along with their total citation counts.

| General Topic | Survey count | Total citations |
|---|---|---|
| Probabilistic methods | 99 | 468 |
| Reinforcement learning | 182 | 3006 |
| Data Governance | 317 | 3787 |
| Data-Driven Applications | 695 | 19539 |
| Machine Learning Optimization functions | 232 | 6327 |
| Modern AI & Data Management | 188 | 6125 |
| General machine learning | 397 | 11394 |
| Low-Level and Efficient Vision | 154 | 2290 |
| Deep Learning | 550 | 23729 |

### 3.1.2 EVALUATION PROMPT PREPARATION

**Content-based Evaluation Prompt.** We design a prompt that explicitly aligns model judgment with human-authored standards. Given a survey topic $T$, a high-quality human-written survey $S^{(h)}$, and an evaluation dimension $d \in \mathcal{D}$ (see Section 3.2.1), the prompt instructs the judge model $\mathcal{M}$ to assess whether an automatically generated survey $S^{(a)}$ satisfies the same quality requirements as $S^{(h)}$. Formally, the prompt is denoted as $P(T, S^{(h)}, S^{(a)}, d, \mathcal{C}_d)$, where $\mathcal{C}_d = \{c_1, c_2, c_3, c_4, c_5\}$ specifies the scoring criteria (levels 1–5). The model $\mathcal{M}$ then outputs a corresponding score (1 to 5).

**Quiz-based Evaluation Prompt.** The evaluation prompt has two components. **(1) Answer Generation** enforces the LLM answers must (i) rely only on RAG-retrieved Gao et al. (2023) passages (Section 3.3.3) without external knowledge, (ii) return exactly "No relevant content found in the survey" if no sufficient information is available, and (iii) list the identifiers of all supporting passages. With this prompt, the RAG-retrieved passages are then provided as reference documents, and the required format specifies that each response include the answer, the supporting passage IDs, and the corresponding source text. **(2) Answer Scoring** guides the LLM evaluator through three tasks: (i) scoring an answer against a ground-truth reference with a predefined rubric for accuracy, completeness, and relevance; (ii) checking whether every claim is directly supported by the provided references (outputting only "True" or "False"); (iii) comparing two answers to the same quiz, judging which is superior in accuracy, completeness, clarity, and helpfulness, with a justification of at most 50 words.

### 3.2 BENCHMARK FEATURES

### 3.2.1 EVALUATION METRICS

For content-based survey evaluation, we evaluate from two key aspects, i.e., outline quality and content quality. First, **outline quality** examines the global organization of the survey. This includes

evaluating whether the outline (1) comprehensively covers key aspects and representative directions of the topic (coverage), (2) maintains topical alignment without off-topic sections (relevance), and (3) reflects a clear and logical hierarchy among sections (structure). Second, for **content quality**, it focuses on the depth and informativeness of the generated text. Specifically, we assess whether each chapter (1) includes key subtopics and representative works (coverage), (2) offers meaningful analysis and synthesis, such as identifying research gaps or future directions (depth), (3) stays centered on its assigned theme (focus), (4) presents ideas in a logically connected and well-structured manner (coherence), and (5) is fluent and grammatically natural (fluency). In addition, we propose a richness metric to quantify the proportion of non-text elements (e.g., charts, diagrams), which is defined as $\text{Richness} = \lambda \cdot \frac{N_{\text{non-text}}}{\sum_{i=1}^{C} L_i}$, where $N_{\text{non-text}}$ denotes the total number of non-text elements (e.g., charts, figures, diagrams), $\sum_{i=1}^{C} L_i$ represents the accumulated length of all $C$ chapters (measured in characters), and $\lambda$ is a tunable hyper-parameter. Notably, we adopt win-rate for quiz-based evaluation.

### 3.2.2 EVALUATION QUIZ SET

We predefine a set of carefully structured quiz (templates) that guide the LLM to evaluate the survey's quality across diverse technical review perspectives and levels.

Table 2: Templates of General Quizzes.

| Difficulty / Num. | Perspective / Num. | Example |
|---|---|---|
| Easy / 10 | Concept Definition / 4 | What is the rigorous definition of {topic}? |
| | Knowledge Classification / 4 | Does {topic} involve any classification of techniques? If so, list the classification criteria and the resulting categories. |
| | Historical Understanding / 2 | List the key stages and evolutionary trajectory of {topic} from its origin to its current state. |
| Medium / 8 | Algorithmic Principles / 2 | Are the main algorithms described in {topic} consistent with the original papers or authoritative sources? |
| | Practical Guidance / 3 | Does {topic} include detailed implementation steps, configurations, parameter selections, or code snippets for its key techniques? |
| | Performance Insights / 3 | For the various techniques involved in {topic}, which performance metrics and evaluation methods does the survey use for each? |
| Hard / 4 | Future Work / 4 | Does the survey provide clear predictions regarding future research directions or technological developments? |

**General Quizzes.** As shown in Table 2, we design a hierarchy of question templates to provide objective, fine-grained evaluation of survey quality. These templates capture the essential characteristics of a high-quality survey. (1) Easy-Level Quizzes test fundamental coverage: *(i) Concept Definition* checks whether key concepts—such as the topic, its motivation, challenges, and related technologies—are clearly and accurately defined, consistent with standard usage; *(ii) Taxonomy* examines the coherence and completeness of taxonomies, including the soundness of classification criteria and the logical flow of resulting structures; *(iii) Historical Context* evaluates whether the survey traces major milestones of the field with accurate, verifiable timelines. (2) Medium-Level Quizzes focus on technical depth: *(i) Algorithmic Principles* assesses the correctness and clarity of core algorithm descriptions and illustrative examples; *(ii) Practical Guidance* checks for implementation details, parameter settings, and real-world usage scenarios; *(iii) Performance Analysis* verifies the use of proper evaluation metrics, clear presentation of results, and reproducible, data-grounded conclusions. (3) Hard-Level Quizzes target higher-order reasoning: *Insights* probe the survey's ability to predict future trends, propose novel ideas, and reason about uncertainties or limitations—reflecting high-level synthesis and forward-looking perspective.

**Topic-Specific Quizzes.** We construct topic-specific quizzes using a RAG-based strategy. Candidate paragraphs are first retrieved from the technical sections of high-quality existing surveys and are verified for informational completeness. To ensure quality, we further filter the candidate paragraphs based on whether they (1) meet a minimal length requirement and (2) pass checks on formula density, media references, sentence completeness, key terminology, and list-like structure. For each retained paragraph, a structured prompt combines the central sentence and full paragraph, instructing the model to generate self-contained quizzes that can be answered solely from the provided text and include accurate, text-grounded answers.

Each generated quiz-answer pair then undergoes multi-criteria validation. Specifically, the quiz must exceed a minimum length; and answers must contain at least two substantive sentences and avoid vague or speculative language while presenting concrete indicators (e.g., numerical data, explicit methods, causal links, ordered discourse markers). Finally, we examine whether the answers remain closely tied to the source paragraphs (keyword-overlap check).

### 3.3 SURVEY EVALUATION

#### 3.3.1 FAIRNESS-GUARANTEED SURVEY WRITING

With our well-prepared dataset (see Section 3.2), we employ LLMs to generate surveys on the selected topics. To avoid potential bias caused by referencing human-written surveys, we explicitly instruct methods like OPENAI-DEEPRESEARCH not to consult existing surveys on relevant topics when generating its outputs (see Appendix C). For rest methods, fairness is naturally ensured, since they are only allowed to access the titles and abstracts of retrieved papers during survey writing.

#### 3.3.2 CONTENT-BASED SURVEY EVALUATION

To ensure rigorous evaluation, we employ a diverse set of methods to assess LLM-generated surveys. The core of our evaluation method is the LLM-as-judge approach, which quantifies outline and content quality using LLMs. This evaluation method consists of two main settings:

**Without Human-Written Surveys as Reference.** In this setting, we evaluate only surveys generated by LLM4Survey methods based on a given topic. More specifically, for content quality evaluation, two evaluation strategies are adopted: (1) **Document-level Evaluation**: The LLM scores the entire generated survey as a whole. (2) **Chapter-level Evaluation**: The LLM scores each paragraph or section individually, and the final score is computed as the average across sections. For chapter-level evaluation, we first average the scores of all sections within a survey, and then take the mean across all topics to obtain the final score for each dimension.

**With Human-Written Surveys as Reference.** Here, we only perform full-document evaluation. The LLM judge is presented with both the LLM-generated survey and a high-quality human-written counterpart. It then assigns a final score based on their relative quality.

#### 3.3.3 QUIZ-BASED SURVEY EVALUATION

Beyond content-based evaluation, we employ "thinking-inspiring" quizzes (Section 3.2.2) to assess surveys without relying on human-written references.

**Retrieval-Augmented Context Selection.** Given a survey, we first extract its hierarchical headings to construct an outline. This outline, along with the quiz, is fed to an LLM (GPT-4o-mini) to identify the most relevant sections, which are retained as candidate context. The remaining text is segmented into paragraphs, and vector similarity is computed between each paragraph and the quiz. Paragraphs with high relevance are selected and paired with their original headings. An optional LLM-based filtering step removes any residual irrelevant content.

**LLM Quiz Answering Process.** Each quiz is paired with the retrieved context to form an LLM prompt. To ensure grounding, the prompt explicitly instructs the LLM to answer solely based on the provided text and to include supporting evidence. This design mitigates hallucination and facilitates downstream verification by enforcing reference-based reasoning.

**LLM Answer Verification and Scoring.** We evaluate both the correctness and evidential grounding of each answer. For general quizzes without gold answers, we prompt an LLM to assess the generated answer based on its cited evidence and assign a quality score on a predefined scale of [0, 10]. For topic-specific quizzes with reference answers, the LLM is additionally provided with the reference but instructed to maintain independent judgment. Crucially, the answer is automatically scored zero if the evidence is deemed insufficient by the LLM, regardless of surface plausibility.

## 4 EXPERIMENTS

We evaluate four typical methods to verify the effectiveness of SurveyBench, including AUTOSURVEY (GPT-4o), SURVEYFORGE (GPT-4o), LLM×MAPREDUCE-V2 (Gemini-flash-thinking), and OPENAI-DEEPRESEARCH (see Section 2).

### 4.1 OVERALL PERFORMANCE RESULTS

We first present preliminary results and observations from content-based and quiz-based evaluation.

Table 3: Content-Based Evaluation Results (w/ human referenced). Note we (1) set the carefully selected human-written surveys scoring **5** in outline and content quality; and (2) test the human-written surveys that score **9.80**, 5.45, **11.68** in the three Richness metrics ($\lambda = 10^5$).

| Dimension | OPENAI-DR | AUTOSURVEY | SURVEYFORGE | LLM×MR-V2 |
|---|---|---|---|---|
| **Outline Quality (1-5)** | | | | |
| Coverage | 3.39 | 3.83 | 3.86 | **4.31** |
| Relevance | 3.83 | 4.11 | 4.22 | **4.53** |
| Structure | 3.48 | 3.71 | 3.95 | **4.28** |
| Average | 3.57 | 3.88 | 4.01 | **4.37** |
| **Content Quality (1-5)** | | | | |
| Coverage | **4.32** | 3.90 | 3.90 | 4.03 |
| Depth | **4.40** | 3.72 | 3.72 | 4.05 |
| Focus | **4.80** | 4.35 | 4.28 | 4.62 |
| Coherence | **4.25** | 3.98 | 4.00 | 4.00 |
| Fluency | **4.32** | 4.20 | 4.25 | 4.30 |
| Average | **4.42** | 4.03 | 4.03 | 4.20 |
| **Richness** | | | | |
| Avg. Fig. Num. | 0.60 | – | 4.10 | – |
| Avg. Table Num. | 0.60 | – | **10.95** | – |
| Total Avg. | 1.78 | – | 5.04 | – |

Table 4: Quiz-Based Evaluation Results.

| Method | General Quiz Template (Win-rate vs human survey) | | | | | | | Topic-Specific Quiz Template (Score:0-10; human survey as 10) |
|---|---|---|---|---|---|---|---|---|
| | Easy | | | Medium | | | Hard | Topic-related details |
| | Concept | Classification | History | Algorithm | Application | Profiling | Prediction | |
| AutoSurvey | 47.4% | 24.6% | 65.4% | 28.0% | 40.0% | 34.5% | 53.3% | 1.58 |
| SurveyForge | 39.7% | 39.1% | 37.9% | 20.0% | 41.5% | 56.7% | 52.5% | 1.48 |
| LLM×MR-V2 | **57.7%** | 48.1% | 50.0% | 36.4% | 48.6% | **60.9%** | 61.9% | **3.19** |
| OpenAI-DR | 53.8% | **55.9%** | **77.5%** | **68.0%** | **69.8%** | 47.3% | **69.2%** | 1.97 |

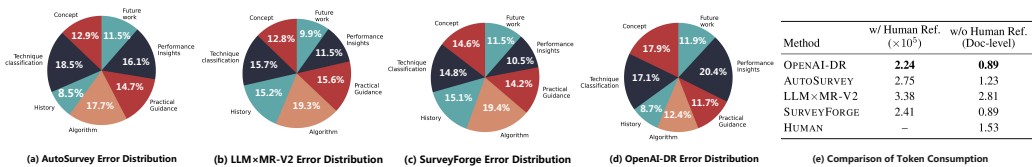

| | (a) AutoSurvey Error Distribution | (b) LLM×MR-V2 Error Distribution | (c) SurveyForge Error Distribution | (d) OpenAI-DR Error Distribution | (e) Comparison of Token Consumption |

| Method | w/ Human Ref. (×10⁵) | w/o Human Ref. (Doc-level) |
|---|---|---|
| OPENAI-DR | **2.24** | **0.89** |
| AUTOSURVEY | 2.75 | 1.23 |
| LLM×MR-V2 | 3.38 | 2.81 |
| SURVEYFORGE | 2.41 | 0.89 |
| HUMAN | – | 1.53 |

Figure 5: Fine-Grained Evaluation. (a-d): Error Distributions; (e) LLM Token Utilization.

**Content-based Evaluation.** Unlike the inflated results observed without human-written references (Tables 7, 8), we incorporate high-quality human-authored surveys as reference standards and instruct the LLM judge to score accordingly. As shown in Table 3, LLM-based methods achieve strong results across these content-focused metrics. For instance, OPENAI-DR scores only 4% below humans in content focus; and LLM×MR-V2 lags by just 9% in outline relevance. That indicates that *LLM-written surveys can approach human surveys in readability and local coherence.* Among these methods, LLM×MR-V2 achieves the highest outline quality, aided by entropy-driven convolutional scaling at test time. OPENAI-DR achieves the highest content quality due to reinforcement learning tailored for complex retrieval. However, OPENAI-DEEPRESEARCH ranks lowest in outline quality, as its outlines remain concise and often omit hierarchical sub-sections.

Additionally, SurveyBench also supports element richness of the generated surveys. As shown in Table 3, human-written surveys score much higher than the LLM-based ones (e.g., ~5.56 times higher than OPENAI-DEEPRESEARCH). The reasons are three-fold. First, methods like SURVEYFORGE and AUTOSURVEY do not provide functionalities for generating diagrams or tables. Second, general agents like OPENAI-DEEPRESEARCH can only produce or use a modest number of figures because they tend to default to generic textual summarization rather than incorporating rich multimodal evidence. Finally, LLM×MR-V2 leverages a templating mechanism to generate images from characters (e.g., Mermaid diagrams) and produces a substantial number of tables, with an average of 10.95 compared to 5.45 for human-written surveys. However, due to its considerably longest outputs among all the methods, its overall richness score remains moderate at 5.04.

Table 5: Evaluation results on outline quality and content quality across *New* vs. *Old* topics.

| Method | Topic Recency | Outline Quality | | | | Content Quality | | | | | |
| | | Coverage | Relevance | Structure | Avg. | Coverage | Depth | Focus | Coherence | Fluency | Avg. |
|---|---|---|---|---|---|---|---|---|---|---|---|
| OPENAI-DR | New | 3.15 | 3.60 | 3.35 | 3.37 | 4.10 | 4.20 | 4.65 | 4.10 | 4.15 | 4.24 |
| | Old | 3.63 | 4.07 | 3.62 | **3.77** | 4.55 | 4.60 | 4.95 | 4.40 | 4.50 | **4.60** |
| AUTOSURVEY | New | 3.67 | 3.98 | 3.55 | 3.73 | 3.75 | 3.55 | 4.15 | 3.90 | 4.25 | 3.92 |
| | Old | 4.00 | 4.23 | 3.87 | **4.03** | 4.05 | 3.90 | 4.55 | 4.05 | 4.15 | **4.14** |
| SURVEYFORGE | New | 3.68 | 4.20 | 3.90 | 3.93 | 3.70 | 3.55 | 4.20 | 3.80 | 4.15 | 3.88 |
| | Old | 4.03 | 4.23 | 4.00 | **4.09** | 4.10 | 3.90 | 4.35 | 4.20 | 4.35 | **4.18** |
| LLM×MR-V2 | New | 4.33 | 4.53 | 4.30 | **4.39** | 3.90 | 3.95 | 4.45 | 4.00 | 4.30 | 4.12 |
| | Old | 4.28 | 4.53 | 4.25 | 4.36 | 4.15 | 4.15 | 4.80 | 4.00 | 4.30 | **4.28** |

**Quiz-based Evaluation.** Next, we conduct a comprehensive assessment using both general quizzes and content-specific quizzes (see Section 3.2.2). The results are reported in Table 4. All four methods yield relatively low scores due to the fine-grained nature of the quizzes, which demand close alignment with the human-written surveys. The findings are summarized as follows.

> **Finding 1:** *Insufficient detail. LLM-generated surveys tend to provide only surface-level explanations of key concepts and techniques and perform poorly in content-specific quizzes.*

For instance, quizzes such as *"What strategies are proposed for adapting instructions into multilingual resources, and how do they differ?"* remain unanswered even by the survey written by OPENAI-DEEPRESEARCH, which lacks the necessary fine-grained discussion, caused by its tendency to remain at a high-level overview without delving into the nuanced differences among concrete methods.

> **Finding 2:** *Lack of associative reasoning. LLM-generated surveys struggle to establish meaningful analogies or cross-concept connections.*

For instance, the quiz *"How does the organizational structure of tiles and pixels relate to CUDA programming architecture?"* evaluates whether a survey can recognize that the handling of tiles and pixels in rendering parallels the blocks and threads in CUDA programming. However, LLM-generated surveys almost entirely omit such associative reasoning, caused by their lack of deep cross-domain coverage and reasoning ability.

> **Finding 3:** *Deficient synthesis and abstraction. LLM-generated surveys seldom offer overarching summaries or integrate key ideas.*

Most quizzes probing main aspects, methods, or dimensions remain unanswered. Because existing methods lack robust capabilities for independent induction, clustering, and summarization. The majority of summary content is directly rewritten from cited sources, without clear self-assessment of importance, relevance, or ordered discussion.

> **Finding 4:** *Pronounced forward-looking content. Despite the above limitations, most LLM-generated surveys consistently include sections on future developments, demonstrating some ability to analyze emerging trends and provide supporting rationale.*

Notably, human authors often tailor the organization and may omit chapters like forward-looking based on individual emphasis. Instead, LLM-based methods adhere to a standardized structural template with remarkable consistency and almost never leave such discussions out.

## 4.2 FINE-GRAINED ANALYSIS

**Error Distributions.** As illustrated in Figure 5 (a-c), there are three main observations. First, OPENAI-DEEPRESEARCH excels in algorithmic principles and structural classification, demonstrating strong technical depth, yet struggles with comparative performance analysis and, at higher technical granularity, shows declining accuracy in conceptual understanding. In contrast, AUTOSURVEY suffers the most from errors in technology-related content, revealing a clear deficiency in both detailed technical knowledge and performance evaluation capabilities. Meanwhile, LLM×MAPREDUCE-V2

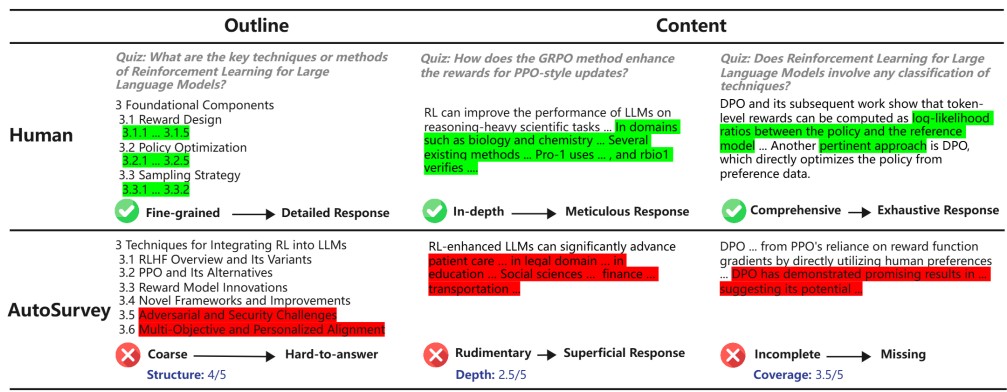

Figure 6: Case Study of Surveys Generated by Human and AUTOSURVEY.

and SURVEYFORGE present nearly identical error distributions, suggesting shared implementation strategies. Though both improve upon performance–insight comprehension, their understanding of deeper algorithmic mechanisms remains shallow.

**Token Consumption.** As shown in Figure 5 (e), OPENAI-DEEPRESEARCH incurs the lowest token usage among all methods, because it produces simple outlines but precise technical analysis. In contrast, LLM×MR-V2 consumes over 33.7% more tokens, as it produces more fine-grained chapter structures and incorporates non-textual elements such as tables.

**Topic Recency.** To examine the effect of topic recency, we sort the 20 evaluation topics according to the release time of the first versions of their corresponding human-written surveys, and divide them equally into 10 *New* and 10 *Old* topics. Table 5 reports the evaluation results across *New* and *Old* topics. Overall, we observe that all methods achieve higher scores on old topics than on new ones, suggesting that topic familiarity contributes positively to the quality of generated surveys. LLM×MAPREDUCE-V2 still exhibits the strongest overall performance, with average outline and content quality scores of **4.39** and **4.29** on old topics, respectively. Similar trends hold for AUTOSURVEY and SURVEYFORGE, though their gains on old topics are less substantial. *These findings highlight that though current methods can already produce competitive surveys on unseen topics, they are more effective when the topic is closer to previously seen or more established domains, partly because older topics are supported by a richer body of literature and a more mature research structure, whereas newer topics have fewer accessible references and less well-formed frameworks.*

## 4.3 CASE STUDY

We conduct a case study on the reinforcement learning (RL) topic by analyzing surveys written by humans and AUTOSURVEY. As illustrated in Figure 6, we highlight key sections that influence metric scores and quiz performance. The human-written survey consistently outperforms AUTOSURVEY, corroborating the quantitative evaluation results. Specifically, for outline structures, the human-written survey provides a fine-grained organization that leads to detailed responses, whereas AUTOSURVEY adopts a coarse structure that often results in hard-to-answer quizzes. For content depth, the human one delivers in-depth and meticulous responses, while AUTOSURVEY remains rudimentary and superficial. And for content coverage, the human one offers comprehensive and exhaustive responses, in contrast to the incomplete and missing coverage in AUTOSURVEY's.

## 5 CONCLUSION

We present SurveyBench, a fine-grained, quiz-driven benchmark for rigorously evaluating automatic academic survey writing. By integrating curated topics, human-aligned scoring, and both content- and quiz-based evaluations, SurveyBench enables comprehensive assessment beyond surface fluency. Empirical results show that while LLM-generated surveys exhibit structural coherence, they fall short in aspects like technical detail, reasoning, and core idea abstraction.

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

## A DETAILS OF TOPICS AND HUMAN SURVEYS

Table 6: Survey Table with Release Time.

| Topic | Survey Title | Release Time | Citations |
|---|---|---|---|
| Graph Neural Networks | Graph Neural Networks: Taxonomy, Advances and Trends | 2020.12 | 203 |
| Vision Transformers | A Survey of Visual Transformers | 2021.11 | 597 |
| 3D Object Detection in Autonomous Driving | 3D Object Detection for Autonomous Driving: A Comprehensive Survey | 2022.06 | 380 |
| Generative Diffusion Models | A Survey on Generative Diffusion Models | 2022.09 | 682 |
| Large Language Models for Recommendation | A Survey on Large Language Models for Recommendation | 2023.05 | 606 |
| Multimodal Large Language Models | A Survey on Multimodal Large Language Models | 2023.06 | 491 |
| Alignment of Large Language Models | Aligning Large Language Models with Human: A Survey | 2023.07 | 438 |
| Evaluation of Large Language Models | A Survey on Evaluation of Large Language Models | 2023.07 | 4073 |
| LLM-based Multi-Agent | A Survey on Large Language Model based Autonomous Agents | 2023.08 | 1903 |
| Hallucination in Large Language Models | Siren's Song in the AI Ocean: A Survey on Hallucination in Large Language Models | 2023.09 | 1465 |
| Explainability for Large Language Models | Explainability for Large Language Models: A Survey | 2023.09 | 890 |
| Retrieval-Augmented Generation for Large Language Models | Retrieval-Augmented Generation for Large Language Models: A Survey | 2023.12 | 3170 |
| 3D Gaussian Splatting | A Survey on 3D Gaussian Splatting | 2024.01 | - |
| Large Language Models for Time Series | Large Language Models for Time Series: A Survey | 2024.02 | 120 |
| Efficient Inference for Large Language Models | A Survey on Efficient Inference for Large Language Models | 2024.04 | 13 |
| Safety in Large Language Models | A Comprehensive Survey in LLM(-Agent) Full Stack Safety: Data, Training and Deployment | 2025.04 | 48 |
| Vision-Language-Action Models | Vision-Language-Action Models: Concepts, Progress, Applications and Challenges | 2025.05 | 24 |
| Scientific Large Language Models | A Survey of Scientific Large Language Models: From Data Foundations to Agent Frontiers | 2025.08 | - |
| Reinforcement Learning for Large Language Models | A Survey of Reinforcement Learning for Large Reasoning Models | 2025.09 | 2 |
| Agentic Reinforcement Learning | The Landscape of Agentic Reinforcement Learning for LLMs: A Survey | 2025.09 | 3 |

## B EXPERIMENTAL RESULTS OF "WITHOUT HUMAN AS REFERENCE"

Table 7: Evaluation results of different methods on outline quality and content quality, without the human-written survey serving as the reference. (Document-level content quality evaluation)

| Method | Outline Quality | | | | Content Quality | | | | | |
|---|---|---|---|---|---|---|---|---|---|---|
| | Coverage | Relevance | Structure | Avg | Coverage | Depth | Focus | Coherence | Fluency | Avg |
| OPENAI-DR (OpenAI, 2025) | 4.77 | 4.99 | 4.79 | 4.85 | 5.00 | 4.97 | 5.00 | 5.00 | 4.88 | 4.97 |
| AUTOSURVEY (Wang et al., 2024) | 4.98 | 5.00 | 4.93 | 4.97 | 5.00 | 5.00 | 5.00 | 5.00 | 4.97 | 5.00 |
| SURVEYFORGE (Yan et al., 2025) | 5.00 | 5.00 | 5.00 | 5.00 | 5.00 | 5.00 | 5.00 | 5.00 | 5.00 | 5.00 |
| LLM×MR-V2 (Wang et al., 2025) | 4.99 | 5.00 | 4.99 | 4.99 | 5.00 | 5.00 | 5.00 | 5.00 | 5.00 | 5.00 |
| HUMAN | 4.90 | 4.99 | 4.91 | 4.93 | 5.00 | 5.00 | 5.00 | 4.97 | 4.70 | 4.94 |

Table 8: Without human survey as reference. (Chapter-level content quality evaluation)

| Method | Content Quality | | | | | |
|---|---|---|---|---|---|---|
| | Coverage | Depth | Focus | Coherence | Fluency | Avg |
| OPENAI-DR (OpenAI, 2025) | 4.97 | 4.66 | 4.99 | 4.81 | 4.63 | 4.81 |
| AUTOSURVEY (Wang et al., 2024) | 4.98 | 4.95 | 4.97 | 4.95 | 4.97 | 4.96 |
| SURVEYFORGE (Yan et al., 2025) | 5.00 | 4.97 | 5.00 | 4.96 | 4.99 | 4.98 |
| LLM×MR-V2 (Wang et al., 2025) | 5.00 | 4.94 | 5.00 | 4.96 | 4.96 | 4.97 |
| HUMAN | 4.88 | 4.60 | 4.98 | 4.51 | 4.38 | 4.67 |

## C OPENAI-DEEPRESEARCH PROMPTS

```
Please write an acdamic survey on the topic of {topic}.

Note: Do not use existing relevant surveys as reference.
```

Figure 7: Initial requirement prompt.

```
Write a comprehensive academic survey on the given topic. The survey should be broad
in coverage, accessible to readers from newcomers to advanced researchers, and well-
structured. Please include: (1) formal definitions of key concepts; (2) diagrams, tables,
or figures where useful; (3) a historical timeline of major milestones; (4) coverage of
foundations, recent methods, open challenges, and applications across related subfields.
The coverage should be as broad as possible while emphasizing the most recent
developments. The writing style should balance technical depth with clarity, making the
content understandable for a wide audience while still rigorous and scholarly. Do not
rely on or reference existing surveys on similar topics; instead, construct the survey
independently based on fundamental sources and reasoning. The length and structure should
follow the conventions of standard academic surveys (e.g., multiple sections, sufficient
depth, and appropriate page count).
```

Figure 8: Further requirement prompt.

# D  WORKFLOW OF TOPIC-SPECIFIC QUIZ GENERATION

---

**Algorithm 1** Enhanced Q–A Generation with Guaranteed Target

---

**Require:** Target number $N$, Questions per segment $k$, Max attempts $m$
**Ensure:** List of $N$ Q–A pairs
1: Initialize *Results* $\leftarrow \emptyset$, *ProcessedSegments* $\leftarrow \emptyset$
2: Set *QualityThreshold* $\leftarrow 0.7$, *Attempts* $\leftarrow 0$
              ▷ Phase 1: Progressive quality degradation
3: **while** $|Results| < N$ **and** *Attempts* $< m$ **do**
4:     *Attempts* $\leftarrow$ *Attempts* $+ 1$
5:     $q \leftarrow N - |Results|$, $s \leftarrow \min(\lceil q/k \rceil, 10)$
              ▷ Dynamic strategy adjustment
6:     **if** *Attempts* $> 0.3m$ or no segments found **then**
7:         *QualityThreshold* $\leftarrow \max(QualityThreshold - 0.1, 0.3)$
8:     **end if**
9:     Sample $s$ segments with *QualityThreshold*
10:     **if** no segments and $|ProcessedSegments| > 0.8 \times Total$ **then**
11:         *ProcessedSegments* $\leftarrow \emptyset$                  ▷ Reset for reuse
12:     **end if**
13:     **for all** segment *seg* not in *ProcessedSegments* **do**
14:         Generate Q–A pairs from *seg* (up to $\min(k, q)$ pairs)
15:         **if** generation succeeds **then**
16:             Add pairs to *Results*, *seg* to *ProcessedSegments*
17:         **end if**
18:         **if** $|Results| \geq N$ **then break**
19:     **end if**
20:     **end for**
21: **end while**
              ▷ Phase 2: Fallback from successful segments
22: **if** $|Results| < N$ **then**
23:     **for all** successful segment from *Results* **do**
24:         Generate additional pairs with relaxed validation
25:         **if** $|Results| \geq N$ **then break**
26:     **end if**
27:     **end for**
28: **end if**
              ▷ Phase 3: Emergency generation
29: **if** $|Results| < 0.8N$ **then**
30:     Generate from any available segments with minimal requirements
31: **end if**
32: **return** first $N$ pairs from *Results*

---

