# OpenReview forum: "SurveyBench: How Well Can LLM(-Agents) Write Academic Surveys?"
_ICLR.cc/2026/Conference — Submitted to ICLR 2026_

### Official Review · Reviewer_gxsR · 2025-10-23

**Soundness:** 3
**Presentation:** 3
**Contribution:** 3
**Rating:** 6
**Confidence:** 4

**Summary:**

**Motivation**: The paper argues that automatic survey writing tools need a reader-aligned, fine-grained benchmark. Existing systems write fluent text but often miss coverage, depth, and usable takeaways. The goal is to evaluate whether LLM agents can produce surveys that people can actually learn from.

**Approach**: The authors build SurveyBench. It selects representative topics from 11,343 recent arXiv papers and pairs them with 4,947 high-quality human surveys. It proposes a hierarchical metric set for outline quality and content quality, plus a richness measure for non-text elements. It adds a dual protocol: content-based judging with and without human references, and quiz-based answerability using RAG and rubric-guided LLM judging. Fairness rules instruct generators not to consult existing surveys.

**Key results**: Across four systems, content-based scores approach human quality on surface metrics, yet quiz-based scores remain low, especially on topic-specific quizzes. LLM×MAPREDUCE-V2 leads outline quality, while OpenAI-DeepResearch leads several content dimensions. Human surveys remain much richer in figures and tables despite one method producing many tables by template. Results are stronger on older topics than new ones.

**Strengths:**

**Task importance**: The benchmark and its survey-writing task directly target the ability to produce reliable scientific surveys. This capability is central for AI4Science, where researchers need concise, trustworthy syntheses to guide experiments, interpret literature, and plan new work.

**Benchmark**: The protocol evaluates both structure and content and anchors scores to answerability, so higher scores reflect surveys that actually teach readers something usable. Careful curation of topics and references, plus validated quizzes, raises difficulty in a controlled way and reduces noise. Broad coverage across subfields reveals strengths and weaknesses that generic writing would miss.

**Evaluation Results**: The findings complement the careful curation and task setup by showing that the protocol reliably separates clean outlines from shallow content, that answerability scores track real comprehension, and that human surveys still lead, which signals clear headroom for improvement. The multi dimensional metrics provide actionable diagnostics for developers.

**Weaknesses:**

**LLM-as-judge validity**: I guess my major concern with this benchmark is in heavy reliance on LLM judges risking bias and instability across seeds or model versions. The paper misses reporting inter-judge agreement, calibration, and confidence intervals for win-rates and content scores. Since the paper heavily uses the LLM-as-judge framework for the evaluation, there should be thorough supporting analysis validating the robustness of LLM evaluation framework.

**Richness metric**: The richness formula rewards more figures or tables per character length. One method can inflate table counts via templating, yet still lags humans in overall richness due to very long outputs. The metric may invite superficial additions instead of evidence-grounded visuals. The authors should discuss any limitations or possible solutions to resolve such issues during evaluation.

**Model leakage**: Instructing agents not to consult existing surveys is hard to enforce, and many human surveys used as references are likely in pretraining corpora. The authors must conduct more analysis (e.g., checking citations or overlap detection) to verify whether the leakage can occur during evaluation.

**Presentation**: Paper’s presentation is generally well written. Though Some tables and figures are crowded (e.g., Figure 5 would benefit from larger text and clearer axis or legend labeling).

**Questions:**

- How are the LLM judges calibrated across runs. Can you report inter-rater reliability and confidence intervals for both content and quiz scores?
- Beyond instructions, what mechanisms ensure agents do not consult existing surveys? Do you check generated citations for survey overlap or run automated overlap detectors?
- How do agents generate figures? Is it fair to evaluate the quality of the generated figures compared to the human gold labels?

---

> ### Author Response · Authors · 2025-11-24
>
> **Dear Reviewer gxsR,**
>
> Thank you for the thoughtful and detailed review. We truly appreciate your recognition of SurveyBench as well as your constructive suggestions, such as LLM-as-judge validity and Model leakage. We hope our responses address your concerns and welcome any further questions or feedback.
>
> ---
>
> > W1 & Q1. LLM-as-judge validity
>
> Thanks for raising this important concern regarding the reliability of LLM-as-judge evaluation.
>
> To address this, we conducted a detailed robustness analysis using APAD (Average Pairwise Absolute Difference) across three independent runs. APAD is defined as:
>
> $$
> \text{APAD}=\frac{2}{n(n-1)}\sum_{i<j}|s_i-s_j|
> $$
>
> where $s_i$ and $s_j$ denote the scores assigned by two different LLM judges, and $n$ is the number of judges. Because APAD quantifies the disagreement between independent runs, a lower APAD reflects higher inter-judge consistency and thus greater robustness. Following your suggestion, we additionally report bootstrapped 95% confidence intervals to quantify uncertainty:
>
> **Content-based Evaluation**
>
> **Content Quality**
>
> | APAD (n = 3) | Coherence | Coverage | Depth  | Fluency | Focus  |
> | ------------ | --------- | -------- | ------ | ------- | ------ |
> | AutoSurvey   | 0.1333    | 0.2500   | 0.3167 | 0.2167  | 0.2500 |
> | LLM×MR-V2    | 0.1000    | 0.1500   | 0.1333 | 0.1833  | 0.2167 |
> | SurveyForge  | 0.1667    | 0.1667   | 0.2500 | 0.2667  | 0.2500 |
> | OpenAI-DR    | 0.1167    | 0.2333   | 0.2500 | 0.1333  | 0.2333 |
>
> | 95% CI      | Coherence        | Coverage         | Depth            | Fluency          | Focus            |
> | ----------- | ---------------- | ---------------- | ---------------- | ---------------- | ---------------- |
> | AutoSurvey  | [3.8750, 4.0083] | [3.8333, 4.0500] | [3.6250, 3.8250] | [4.1917, 4.3333] | [4.0417, 4.2833] |
> | LLM×MR-V2   | [3.9667, 4.0500] | [3.9833, 4.1000] | [4.0083, 4.1333] | [4.2250, 4.3750] | [4.4750, 4.6917] |
> | SurveyForge | [3.8667, 4.0333] | [3.8083, 3.9833] | [3.6417, 3.8583] | [4.1500, 4.3167] | [4.1833, 4.4083] |
> | OpenAI-DR   | [4.2000, 4.3917] | [4.2167, 4.4417] | [4.1917, 4.4417] | [4.2583, 4.4667] | [4.7083, 4.8583] |
>
> **Outline Quality**
>
> | APAD (n = 3) | Coverage | Relevance | Structure |
> | ------------ | -------- | --------- | --------- |
> | AutoSurvey   | 0.0833   | 0.0500    | 0.1333    |
> | LLM×MR-V2    | 0.0667   | 0.0500    | 0.0167    |
> | SurveyForge  | 0.1000   | 0.1000    | 0.1167    |
> | OpenAI-DR    | 0.1500   | 0.2000    | 0.1167    |
>
> | 95% CI      | Coverage         | Relevance        | Structure        |
> | ----------- | ---------------- | ---------------- | ---------------- |
> | AutoSurvey  | [3.7000, 3.9500] | [4.0250, 4.2250] | [3.7167, 3.9083] |
> | LLM×MR-V2   | [4.1833, 4.4167] | [4.4750, 4.5500] | [4.1667, 4.2917] |
> | SurveyForge | [3.8917, 4.0417] | [4.1417, 4.2917] | [3.9417, 4.0417] |
> | OpenAI-DR   | [3.2167, 3.5500] | [3.7833, 4.0333] | [3.3583, 3.6333] |
>
> (To be continued)

---

> ### Author Response · Authors · 2025-11-24
>
> (continued)
>
> **Quiz-based Evaluation**
>
> To minimize the use of the LLM’s prior knowledge during quiz evaluation, we require it to provide a "reference" from the paragraph content when answering each question. When assessing answer correctness, the LLM judge first evaluates the relevance of the reference, assigning a score of zero if the cited content does not support the answer. This procedure helps reduce the influence of the model’s prior knowledge on the evaluation.
>
> | APAD(Winrate, n = 3) | Concept | Classification | History | Alogrithm | Application | Profiling | Prediction | Topic Specific Score |
> | -------------------- | ------- | -------------- | ------- | --------- | ----------- | --------- | ---------- | -------------------- |
> | AutoSurvey           | 0.055   | 0.045          | 0.116   | 0.084     | 0.088       | 0.041     | 0.015      | 0.040                |
> | LLM×MR-V2            | 0.031   | 0.019          | 0.015   | 0.147     | 0.083       | 0.051     | 0.037      | 0.013                |
> | SurveyForge          | 0.043   | 0.103          | 0.099   | 0.074     | 0.091       | 0.045     | 0.033      | 0.013                |
> | DeepResearch         | 0.043   | 0.078          | 0.035   | 0.043     | 0.082       | 0.045     | 0.059      | 0.013                |
>
> | 95% CI       | Concept          | Classification   | History          | Alogrithm         | Application      | Profiling        | Prediction       | Topic Specific Score |
> | ------------ | ---------------- | ---------------- | ---------------- | ----------------- | ---------------- | ---------------- | ---------------- | -------------------- |
> | AutoSurvey   | [0.3333, 0.5387] | [0.1712, 0.3595] | [0.3081, 0.7812] | [0.0282, 0.3725]  | [0.1597, 0.4936] | [0.2996, 0.4684] | [0.4952, 0.5501] | [1.4755, 1.6245]     |
> | LLM×MR-V2    | [0.4966, 0.6287] | [0.4612, 0.5322] | [0.4758, 0.5389] | [-0.0381, 0.5208] | [0.2687, 0.6126] | [0.4766, 0.6654] | [0.5730, 0.7130] | [3.1752, 3.2248]     |
> | SurveyForge  | [0.2721, 0.4432] | [0.1261, 0.5266] | [0.1049, 0.4838] | [0.0698, 0.3482]  | [0.2557, 0.5970] | [0.4262, 0.6184] | [0.4696, 0.5957] | [1.4452, 1.4948]     |
> | DeepResearch | [0.4809, 0.6491] | [0.3591, 0.6602] | [0.7156, 0.8537] | [0.5531, 0.7269]  | [0.4804, 0.7869] | [0.3908, 0.5572] | [0.6297, 0.8550] | [1.9352, 1.9848]     |
>
> **Summary of Findings**
>
> From the results above, we can observe that:
>
> 1. **Content-based evaluation shows strong stability.** APAD values are mostly below 0.3, and the 95% confidence intervals are narrow across all dimensions.
> 2. **Quiz-based evaluation also exhibits high consistency.** APAD values are mostly below 0.1, with small confidence intervals despite the fine-grained nature of the quizzes.
> 3. Together, these results demonstrate that the **SurveyBench LLM evaluation framework is stable and reliable**, even when evaluated across multiple independent runs.

---

> ### Author Response · Authors · 2025-11-24
>
> > W2. Richness metric
>
> Thank you for the insightful suggestion.
>
> We agree that the metric may be susceptible to superficial inflation. Our richness metric is designed to measure the proportion of non-textual elements within a generated survey. As shown in Table 3, this metric effectively reveals a key weakness of current LLM-generated surveys, LLMs produce substantially fewer visual elements than human-written surveys. For example, the richness score of human surveys is approximately 2.3× higher than that of the best-performing LLM-based method, highlighting a gap in visual aid.
>
> One potential direction for future direction is to combine richness with semantic or multimodal verification, using vision–language models to assess whether visuals are meaningful and grounded.
>
> ---
>
> > Q3. Figure comparison fairness
>
> Thanks for your valuable comment.
>
> LLM agents typically generate figures using text-based template languages (e.g., Mermaid). These formats allow models to produce structural diagrams, flowcharts and taxonomies, but they remain far less expressive and polished than expert-crafted figures in human-written high-quality surveys.
>
> Regarding fairness, our richness evaluation does not require LLM-generated figures to match human figures in correctness or complexity. The richness metric only evaluates: (1) whether visual elements are present, and (2) how frequently they appear relative to the overall document length. It does not judge whether the model-produced visuals resemble human gold labels. Instead, human surveys are used merely as reference standards to establish typical expectations for visual density in academic survey writing.
>
> ---
>
> > W3 & Q2. Model leakage
>
> Thanks for your thoughtful suggestion on model leakage.
>
> As described in our paper, we explicitly instruct OpenAI-DR not to reference or consult any existing survey papers. For the remaining methods, the LLM agents are restricted to accessing only the titles and abstracts of retrieved papers. Full survey texts are never provided to the model, which naturally prevents direct leakage of existing surveys.
>
> Following your suggestion, we additionally examine whether generated surveys cite other human-written surveys. Specifically, we compute the proportion of survey citations among all references produced by each method. The results are shown below:
>
> | Method      | Survey Ratio |
> | ----------- | ------------ |
> | OpenAI-DR   | 0.052        |
> | AutoSurvey  | 0.138        |
> | SurveyForge | 0.147        |
> | LLM×MR-V2   | 0.046        |
>
> These ratios are consistently low, indicating that direct survey leakage is minimal across methods.
>
> ---
>
> > W4. Crowded tables and figures
>
> Thank you for your careful inspection and helpful suggestion.
>
> We agree that Figure 5 can be improved for readability. In the final version, we will increase the font size of the text in the pie charts and enhance the clarity of the legends to ensure that all elements are easy to interpret.

---

> ### Comment · Reviewer_gxsR · 2025-11-25
> **Response to Reviewers**
>
> Thank you for the response. The response has addressed some of my concerns; however, given the remaining issue of clearly distinguishing this work from related prior work (raised in other reviews), I will maintain my current score.

---

> > ### Author Response · Authors · 2025-11-26
> >
> > **Dear Reviewer gxsR,**
> >
> > Thank you for the follow-up. We appreciate your comments and fully understand the concern regarding how our work distinguishes itself from related prior work. To present the differences more clearly, we provide the following comparative table, highlighting where SurveyBench introduces novel capabilities that prior benchmarks do not support:
> >
> > | **Bench**              | **Topic Preparation**            | **Outline Quality** | **Content Quality** | **Reference Quality** | **Non-text Metric** | **Multi-setting Evaluation** | **Quiz-based Evaluation** |
> > | ---------------------- | -------------------------------- | ------------------- | ------------------- | --------------------- | ------------------- | ---------------------------- | ------------------------- |
> > | AutoSurvey[1]          | ×                                | ×                   | √ (3 sub-dims)      | √                     | ×                   | ×                            | ×                         |
> > | SurveyForge[2]         | ×                                | √ (1 dim)           | √ (3 sub-dims)      | √                     | ×                   | ×                            | ×                         |
> > | LLM×MR-V2[3]           | ×                                | ×                   | √ (4 sub-dims)      | √                     | ×                   | ×                            | ×                         |
> > | **SurveyBench (Ours)** | **√ (Embedding-based curation)** | **√ (3 sub-dims)**  | **√ (5 sub-dims)**  | ×                     | **√ (Richness)**    | **√ (Dual-setting)**         | **√ (General+Specific)**  |
> >
> > As shown, SurveyBench differs from existing works in several key aspects:
> >
> > - **A rigorous topic-preparation pipeline**
> > - **More fine-grained scoring dimensions (3 outline + 5 content vs. ≤4 in prior work)**
> > - **Dual evaluation settings (with / without human reference)**
> > - **First benchmark to evaluate visual richness (figures/tables)**
> > - **First to introduce quiz-based, answerability-centered evaluation**
> >
> > We hope this clarifies how our benchmark introduces new evaluation perspectives and capabilities not present in prior work, and we thank the reviewer again for the constructive feedback.
> >
> > ---
> >
> > [1] Wang Y, Guo Q, Yao W, et al. Autosurvey: Large language models can automatically write surveys[J]. Advances in neural information processing systems, 2024, 37: 115119-115145.
> >
> > [2] Yan X, Feng S, Yuan J, et al. Surveyforge: On the outline heuristics, memory-driven generation, and multi-dimensional evaluation for automated survey writing[C]//Proceedings of the 63rd Annual Meeting of the Association for Computational Linguistics (Volume 1: Long Papers). 2025: 12444-12465.
> >
> > [3] Wang H, Fu Y, Zhang Z, et al. LLM $\times$ MapReduce-V2: Entropy-Driven Convolutional Test-Time Scaling for Generating Long-Form Articles from Extremely Long Resources[J]. arXiv preprint arXiv:2504.05732, 2025.

---

### Official Review · Reviewer_Mxrn · 2025-10-29

**Soundness:** 1
**Presentation:** 2
**Contribution:** 1
**Rating:** 2
**Confidence:** 4

**Summary:**

This paper focuses on the evaluation of the automatic survey generation. The authors point out that existing evaluation metrics cannot capture the core value of a high-quality survey, such as logical structure, content depth, key insights, and practical utility for readers and propose an evaluation system to comprehensively evaluate the quality of a survey from content, outline. and richness. Experiments demonstrate that SurveyBench can reveal the gaps between AI-generated surveys and surveys by human experts.

**Strengths:**

The motivation of this article is meaningful; a better evaluation of AI-generated surveys could help improve the quality of auto-survey-gen and further improve the efficiency of human research.

**Weaknesses:**

### **Serious Overlap with Prior Work**

**The core contribution of this paper, named SurveyBench, is highly similar to the benchmark with the same name proposed in a previously published paper, SurveyForge [A], which has been published on ACL 2025, from the following perspectives:**

- Core evaluation dimensions overlap: Both go beyond simple text fluency assessment and consider key elements of academic reviews:
  1. Outline Quality: Both evaluate whether the structure of the review is reasonable and the logic is coherent.
  2. Content Quality: Both evaluate whether the content is comprehensive, accurate, and profound.
  3. Reference Quality (richness in this article): Both emphasize the importance of citing key, high-impact literature within the field.

- Survey topic: SurveyBench includes 20 topics, of which 10 are completely identical to those in the published SurveyForge. More seriously, for these 10 overlapping topics, the "human-written survey references" selected as the gold standard in this paper are also exactly the same as those used in the SurveyForge paper.

[A] Yan X, Feng S, Yuan J, et al. Surveyforge: On the outline heuristics, memory-driven generation, and multi-dimensional evaluation for automated survey writing[J]. arXiv preprint arXiv:2503.04629, 2025.

The authors need to clarify the originality and workload of the proposed surveybench. If the originality and workload are sufficient, I will reconsider the score.

**Questions:**

Please refer to the weakness.

**Details Of Ethics Concerns:**

The benchmark is similar to the surveybench in the published paper (Surveyforge: On the outline heuristics, memory-driven generation, and multi-dimensional evaluation for automated survey writing, ACL 2025)

---

> ### Author Response · Authors · 2025-11-24
>
> **Dear reviewer Mxrn,**
>
> We sincerely thank the reviewer for the careful reading and for pointing out potential overlap with SurveyForge. We address your points in detail below and clarify the originality, differences and workload of our SurveyBench framework.
>
> ---
>
> ### **(1) Dimension Overlap**
>
> The dimensions "outline quality" and "content quality" are commonly adopted in both the SurveyForge[1] and other LLM-based survey generation studies (such as AutoSurvey[2], SurveyX[3]). However, the corresponding evaluation metrics in these works are overly generic and they overlook other dimensions necessary for assessing survey quality. In contrast, our work introduces contributions along three key axes:
>
> - **Different sub-dimension design.**
>   SurveyForge evaluates only four dimensions (outline, content structure, content relevance, and content coverage). In contrast, we emphasize fine-grained assessment by decomposing outline quality into 3 sub-dimensions, and content quality into 5 sub-dimensions, allowing deeper diagnostic power during evaluation.
> - **Dual-level content scoring.**
>   Our content quality pipeline includes **both document-level and chapter-level** evaluations, whereas SurveyForge evaluates content only at the full-document level.
> - **Two evaluation settings for both outline and content quality.**
>   We further evaluate each dimension **with and without human-written surveys as reference**, in order to examine how the presence of a human gold standard affects LLM-as-a-judge scoring behavior. SurveyForge does not analyze this effect.
>
> Regarding the reviewer’s concern that our “richness” metric resembles SurveyForge’s “reference quality,” we clarify that they measure **entirely different aspects**.
>
> - SurveyForge’s reference quality evaluates whether the generated survey appropriately cites correct and relevant literature.
> - In contrast, **our richness metric quantifies the density of figures, tables, and diagrams in the generated survey**. This metric captures an important characteristic of human-written surveys—the visual explanatory structure—which has not been quantified in any prior benchmark.
>
> Thus, richness is distinct, novel, and conceptually unrelated to SurveyForge’s reference quality metric.
>
> ---
>
> ### **(2) Survey Topics**
>
> We thank the reviewer for noting the topic overlap. We deliberately include several mature, literature-rich topics also used in SurveyForge to maintain benchmark representativeness and support fair comparison with prior work. To avoid misunderstanding, we will revise our paper to explicitly state that part of our topics are adopted from SurveyForge and add proper citation to SurveyForge.
>
> In addition to the topics involved in SurveyForge, we further expand our benchmark by incorporating a broader set of high-impact and emerging domains. To construct a high-quality and representative topic set, we employ a rigorous topic-preparation pipeline that includes: (1) multi-field embedding clustering using title, abstract, and topic embeddings; (2) pruning based on topic maturity, diversity, and diagnostic value; and (3) **sampling from 11,343 recent arXiv papers and 4,947 curated human-written surveys**. This systematic process ensures that our selected topics are not only diverse but also representative.
>
> (To be continued)

---

> ### Author Response · Authors · 2025-11-24
>
> (continued)
>
> ### **(3) Innovation and Workload of SurveyBench**
>
> SurveyBench introduces multiple innovations and requires substantial effort that differentiate it clearly from SurveyForge and prior work:
>
> 1. **Two-tier quiz-based evaluation (general + topic-specific)**
>    We are the first to introduce a quiz-driven evaluation framework that assesses whether a generated survey can actually answer reader-oriented questions. This includes:
>
>    - general quizzes reflecting core elements of high-quality surveys (concept definition, taxonomy, algorithmic principles, performance insights, future work).
>      - These quizzes were designed by analyzing and distilling insights from highly cited surveys, identifying the high-information-density sections as the key elements of exemplary surveys, and integrating them into the quiz format.
>      - Moreover, to mitigate potential biases and instability of LLM-as-judge in evaluating abstract or difficult-to-quantify aspects like article structure or conceptual coherence, the quiz deliberately focuses on content-level assessments. By framing questions around objectively identifiable elements such as topic taxonomies, the field’s historical development, and major methodological branches, the quiz can indirectly capture the structural soundness and narrative quality of the survey.
>    - topic-specific quizzes automatically generated from validated high-quality paragraphs via a multi-stage filtering and QA pipeline.
>      - These Topic-specific Quiz template assesses content gaps between LLM-generated and human surveys through hierarchical quality control.
>      - Question–answer pairs are generated in three stages with progressively relaxed standards. All pairs must be sufficiently long, context-independent, substantive, non-speculative, and retain partial keyword overlap with the source. Deduplication and paragraph-pool refresh ensure content diversity and robustness.
>        SurveyForge does not include any quiz-based, reader-aligned, or answerability-based evaluation.
>
> 2. **Fine-grained content-based evaluation with a novel multimodal richness metric**
>    SurveyBench offers substantially more granular content-based evaluation than SurveyForge through (i) document-level and chapter-level scoring, (ii) both with- and without-human-reference settings, and (iii) three outline sub-dimensions and five content sub-dimensions that enable detailed diagnostic analysis. Moreover, we introduce a novel richness metric that quantifies the use of non-textual elements—figures, tables, and diagrams—an essential component of human-written surveys that has never been evaluated in prior benchmarks, including SurveyForge.
> 3. **Effectiveness of SurveyBench**
>    Through the clustering-based topic preparation pipeline described in (2), we ensure that the selected topics are both representative and diverse, covering mature, emerging and structurally distinct areas. This results in a well-grounded and diagnostically rich topic set, forming the first benchmark specifically tailored for evaluating academic survey writing.
>    Leveraging this high-quality dataset together with our fine-grained, multi-level, and quiz-based evaluation framework, SurveyBench uncovers several previously unreported findings about the limitations of current LLMs in survey generation, including:
>
>    - LLMs’ systematic weaknesses in algorithmic principles, performance insights, and associative reasoning;
>    - The lack of synthesis and abstraction, where LLMs mainly rewrite retrieved content rather than integrating or prioritizing it;
>    - Strong dependence on topic recency, with significantly lower performance on emerging domains.
>      These insights have not been discussed in prior work and are made possible only through SurveyBench’s uniquely fine-grained and quiz-based evaluation design.
>
> ---
>
> Taken together, these contributions demonstrate that SurveyBench is **a significantly more comprehensive, reader-aligned, and diagnostically rich benchmark**, bringing new evaluation perspectives and infrastructure that do not exist in SurveyForge or any prior work.
>
> [1] Yan X, Feng S, Yuan J, et al. Surveyforge: On the outline heuristics, memory-driven generation, and multi-dimensional evaluation for automated survey writing[C]//Proceedings of the 63rd Annual Meeting of the Association for Computational Linguistics (Volume 1: Long Papers). 2025: 12444-12465.
>
> [2] Wang Y, Guo Q, Yao W, et al. Autosurvey: Large language models can automatically write surveys[J]. Advances in neural information processing systems, 2024, 37: 115119-115145.
>
> [3] Liang X, Yang J, Wang Y, et al. Surveyx: Academic survey automation via large language models[J]. arXiv preprint arXiv:2502.14776, 2025.

---

> ### Author Response · Authors · 2025-11-26
>
> **Dear Reviewer Mxrn,**
>
> Thank you again for raising the important point regarding potential overlap with SurveyForge. Following our previous clarification, we provide a concise comparison to more clearly illustrate how SurveyBench differs from prior work. The table below highlights where SurveyBench introduces capabilities that existing benchmarks do not support:
>
> | **Bench**              | **Topic Preparation**            | **Outline Quality** | **Content Quality** | **Reference Quality** | **Non-text Metric** | **Multi-setting Evaluation** | **Quiz-based Evaluation** |
> | ---------------------- | -------------------------------- | ------------------- | ------------------- | --------------------- | ------------------- | ---------------------------- | ------------------------- |
> | AutoSurvey[1]          | ×                                | ×                   | √ (3 sub-dims)      | √                     | ×                   | ×                            | ×                         |
> | SurveyForge[2]         | ×                                | √ (1 dim)           | √ (3 sub-dims)      | √                     | ×                   | ×                            | ×                         |
> | LLM×MR-V2[3]           | ×                                | ×                   | √ (4 sub-dims)      | √                     | ×                   | ×                            | ×                         |
> | **SurveyBench (Ours)** | **√ (Embedding-based curation)** | **√ (3 sub-dims)**  | **√ (5 sub-dims)**  | ×                     | **√ (Richness)**    | **√ (Dual-setting)**         | **√ (General+Specific)**  |
>
> As shown, SurveyBench differs from existing works in several key aspects:
>
> - **A rigorous topic-preparation pipeline**
> - **More fine-grained scoring dimensions (3 outline + 5 content vs. ≤4 in prior work)**
> - **Dual evaluation settings (with / without human reference)**
> - **First benchmark to evaluate visual richness (figures/tables)**
> - **First to introduce quiz-based, answerability-centered evaluation**
>
> We hope this clarifies how SurveyBench advances beyond prior work, and we welcome any further questions or feedback.
>
> ---
>
> [1] Wang Y, Guo Q, Yao W, et al. Autosurvey: Large language models can automatically write surveys[J]. Advances in neural information processing systems, 2024, 37: 115119-115145.
>
> [2] Yan X, Feng S, Yuan J, et al. Surveyforge: On the outline heuristics, memory-driven generation, and multi-dimensional evaluation for automated survey writing[C]//Proceedings of the 63rd Annual Meeting of the Association for Computational Linguistics (Volume 1: Long Papers). 2025: 12444-12465.
>
> [3] Wang H, Fu Y, Zhang Z, et al. LLM $\times$ MapReduce-V2: Entropy-Driven Convolutional Test-Time Scaling for Generating Long-Form Articles from Extremely Long Resources[J]. arXiv preprint arXiv:2504.05732, 2025.

---

### Official Review · Reviewer_7uCU · 2025-11-01

**Soundness:** 2
**Presentation:** 3
**Contribution:** 3
**Rating:** 4
**Confidence:** 4

**Summary:**

This paper introduces SurveyBench, a benchmark for evaluating LLM-generated academic surveys through fine-grained, quiz-driven assessments. The framework integrates a curated dataset of 20 CS topics, dual evaluation modes (content-based and quiz-based), and hierarchical quality metrics. Experiments on existing systems like DeepResearch and AutoSurvey reveal that, while LLM-generated surveys are structurally coherent, they substantially lag behind human-written ones in content richness and quiz performance. Overall, the work addresses a timely gap in automatic survey evaluation and provides a useful foundation for improving LLM4Survey research.

**Strengths:**

1. This work proposes a dual-mode protocol: content-based evaluation (with human surveys as gold standards) and quiz-driven evaluation (hierarchical general quizzes + RAG-based topic-specific quizzes). This directly targets readers’ informational needs (e.g., technical details, cross-concept reasoning) and fills the gap of “reader-centric assessment” in existing benchmarks
2. The benchmark is curated rigorously: 20 representative CS topics from 11,343 recent arXiv papers and 4,947 high-quality human surveys, filtered via embedding clustering and citation impact. It also builds a hierarchical metric system (outline/content quality, non-textual richness) with clear criteria and bias mitigation (e.g., prohibiting LLM surveys from referencing human ones), ensuring reproducibility
3. Practical in-depth experimental insights It evaluates 4 mainstream LLM4Survey methods, with fine-grained analysis and quantitative gap characterization. These findings identify key LLM weaknesses (insufficient detail, weak associative reasoning) and guide future optimization.

**Weaknesses:**

1. The content-based evaluation heavily relies on the LLM-as-judge approach for scoring outline and content quality (e.g., Coverage, Depth, Focus, Fluency, etc.), which exhibits limitations in discrimination and stability
2. The benchmark's final selection of a limited number of topics presents a significant scale limitation that hinders the comprehensive evaluation of LLM-Agents' generalizability and exposes a risk of selection bias.

**Questions:**

1.	Given that SurveyBench in the paper only covers 20 typical topics in the computer science field, will the narrow coverage of topics make it easy for models to game the system on these topics through targeted training?
2.	Although the paper conducts experiments on topic recency, it remains unclear whether distinct topic types (rather than merely the recency difference) would lead to significant variations in the performance of LLMs when generating survey reports?

---

> ### Author Response · Authors · 2025-11-24
>
> **Dear Reviewer 7uCU,**
>
> Thank you for the thoughtful and detailed review. We truly appreciate your recognition of SurveyBench as well as your constructive suggestions, such as the LLM-as-judge stability and the influence of topic types. We hope our responses address your concerns and welcome any further questions or feedback.
>
> ---
>
> > W1. LLM-as-judge reliability
>
> Thanks for pointing this out. We fully agree that the reliability of LLM-based evaluation must be carefully validated, and we have proposed two complementary techniques to ensure robustness.
>
> **(1) Using human survey references improves discrimination and scoring stability.**
>
> When human-written surveys are provided as references, the LLM judges gain a clearer comparison target, which leads to more consistent and discriminative scoring. This setting helps the LLM judge anchor its scoring based on concrete human standards.
>
> **(2) Robustness validation via APAD and confidence intervals.**
>
> Moreover, we conducted a detailed robustness analysis using APAD (Average Pairwise Absolute Difference) across several independent runs. APAD is defined as:
>
> $$
> \text{APAD}=\frac{2}{n(n-1)}\sum_{i<j}|s_i-s_j|
> $$
>
> where $s_i$ and $s_j$ denote the scores assigned by two different LLM judges, and $n$ is the number of judges. Because APAD quantifies the disagreement between independent runs, a lower APAD reflects higher inter-judge consistency and thus greater robustness. Following your suggestion, we additionally report bootstrapped 95% confidence intervals to quantify uncertainty:
>
> **Content-based Evaluation**
>
> **Content Quality**
>
> | APAD (n = 3) | Coherence | Coverage | Depth  | Fluency | Focus  |
> | ------------ | --------- | -------- | ------ | ------- | ------ |
> | AutoSurvey   | 0.1333    | 0.2500   | 0.3167 | 0.2167  | 0.2500 |
> | LLM×MR-V2    | 0.1000    | 0.1500   | 0.1333 | 0.1833  | 0.2167 |
> | SurveyForge  | 0.1667    | 0.1667   | 0.2500 | 0.2667  | 0.2500 |
> | OpenAI-DR    | 0.1167    | 0.2333   | 0.2500 | 0.1333  | 0.2333 |
>
> | 95% CI      | Coherence        | Coverage         | Depth            | Fluency          | Focus            |
> | ----------- | ---------------- | ---------------- | ---------------- | ---------------- | ---------------- |
> | AutoSurvey  | [3.8750, 4.0083] | [3.8333, 4.0500] | [3.6250, 3.8250] | [4.1917, 4.3333] | [4.0417, 4.2833] |
> | LLM×MR-V2   | [3.9667, 4.0500] | [3.9833, 4.1000] | [4.0083, 4.1333] | [4.2250, 4.3750] | [4.4750, 4.6917] |
> | SurveyForge | [3.8667, 4.0333] | [3.8083, 3.9833] | [3.6417, 3.8583] | [4.1500, 4.3167] | [4.1833, 4.4083] |
> | OpenAI-DR   | [4.2000, 4.3917] | [4.2167, 4.4417] | [4.1917, 4.4417] | [4.2583, 4.4667] | [4.7083, 4.8583] |
>
> **Outline Quality**
>
> | APAD (n = 3) | Coverage | Relevance | Structure |
> | ------------ | -------- | --------- | --------- |
> | AutoSurvey   | 0.0833   | 0.0500    | 0.1333    |
> | LLM×MR-V2    | 0.0667   | 0.0500    | 0.0167    |
> | SurveyForge  | 0.1000   | 0.1000    | 0.1167    |
> | OpenAI-DR    | 0.1500   | 0.2000    | 0.1167    |
>
> | 95% CI      | Coverage         | Relevance        | Structure        |
> | ----------- | ---------------- | ---------------- | ---------------- |
> | AutoSurvey  | [3.7000, 3.9500] | [4.0250, 4.2250] | [3.7167, 3.9083] |
> | LLM×MR-V2   | [4.1833, 4.4167] | [4.4750, 4.5500] | [4.1667, 4.2917] |
> | SurveyForge | [3.8917, 4.0417] | [4.1417, 4.2917] | [3.9417, 4.0417] |
> | OpenAI-DR   | [3.2167, 3.5500] | [3.7833, 4.0333] | [3.3583, 3.6333] |
>
> (To be continued)

---

> ### Author Response · Authors · 2025-11-24
>
> (continued)
>
> **Quiz-based Evaluation**
>
> To minimize the use of the LLM’s prior knowledge during quiz evaluation, we require it to provide a "reference" from the paragraph content when answering each question. When assessing answer correctness, the LLM judge first evaluates the relevance of the reference, assigning a score of zero if the cited content does not support the answer. This procedure helps reduce the influence of the model’s prior knowledge on the evaluation.
>
> | APAD(Winrate, n = 3) | Concept | Classification | History | Alogrithm | Application | Profiling | Prediction | Topic Specific Score |
> | -------------------- | ------- | -------------- | ------- | --------- | ----------- | --------- | ---------- | -------------------- |
> | AutoSurvey           | 0.055   | 0.045          | 0.116   | 0.084     | 0.088       | 0.041     | 0.015      | 0.040                |
> | LLM×MR-V2            | 0.031   | 0.019          | 0.015   | 0.147     | 0.083       | 0.051     | 0.037      | 0.013                |
> | SurveyForge          | 0.043   | 0.103          | 0.099   | 0.074     | 0.091       | 0.045     | 0.033      | 0.013                |
> | OpenAI-DR            | 0.043   | 0.078          | 0.035   | 0.043     | 0.082       | 0.045     | 0.059      | 0.013                |
>
> | 95% CI       | Concept          | Classification   | History          | Alogrithm         | Application      | Profiling        | Prediction       | Topic Specific Score |
> | ------------ | ---------------- | ---------------- | ---------------- | ----------------- | ---------------- | ---------------- | ---------------- | -------------------- |
> | AutoSurvey   | [0.3333, 0.5387] | [0.1712, 0.3595] | [0.3081, 0.7812] | [0.0282, 0.3725]  | [0.1597, 0.4936] | [0.2996, 0.4684] | [0.4952, 0.5501] | [1.4755, 1.6245]     |
> | LLM×MR-V2    | [0.4966, 0.6287] | [0.4612, 0.5322] | [0.4758, 0.5389] | [-0.0381, 0.5208] | [0.2687, 0.6126] | [0.4766, 0.6654] | [0.5730, 0.7130] | [3.1752, 3.2248]     |
> | SurveyForge  | [0.2721, 0.4432] | [0.1261, 0.5266] | [0.1049, 0.4838] | [0.0698, 0.3482]  | [0.2557, 0.5970] | [0.4262, 0.6184] | [0.4696, 0.5957] | [1.4452, 1.4948]     |
> | OpenAI-DR    | [0.4809, 0.6491] | [0.3591, 0.6602] | [0.7156, 0.8537] | [0.5531, 0.7269]  | [0.4804, 0.7869] | [0.3908, 0.5572] | [0.6297, 0.8550] | [1.9352, 1.9848]     |
>
> **Summary of Findings**
>
> As shown in the table above, we can observe that:
>
> 1. **Content-based evaluation shows strong stability.** APAD values are mostly below 0.3, and the 95% confidence intervals are narrow across all dimensions.
> 2. **Quiz-based evaluation also exhibits high consistency.** APAD values are mostly below 0.1, with small confidence intervals despite the fine-grained nature of the quizzes.
> 3. Together, these results demonstrate that the **SurveyBench LLM evaluation framework is stable and reliable**, even when evaluated across multiple independent runs.

---

> ### Author Response · Authors · 2025-11-24
>
> > W2 & Q1. Limited topic field
>
> Thanks for your thoughtful comment.
>
> To address this concern, we additionally expand SurveyBench with 10 topics and their corresponding human survey (such as [1, 2, 3]) from non-CS disciplines, including **representative areas from Chemistry, Materials Science, Mathematics, Physics, Biology, and Medicine.** These topics were selected to cover different writing conventions, structural norms, and reasoning styles commonly seen in natural sciences, thereby testing whether SurveyBench generalize beyond computer science.
>
> Below we report the extended evaluation results.
>
> **Content-based Evaluation**
>
> | Dimenson            | OpenAI-DR | AutoSurvey | SurveyForge | LLM×MR-V2 |
> | ------------------- | --------- | ---------- | ----------- | --------- |
> | **Outline Quality** |
> | Coverage            | 4.18      | 4.27       | 4.60        | **4.67**  |
> | Relevance           | 4.45      | 4.42       | 4.67        | **4.68**  |
> | Structure           | 4.05      | 3.98       | 4.40        | **4.45**  |
> | Average             | 4.23      | 4.22       | 4.56        | **4.60**  |
> | **Content Quality** |
> | Coverage            | **4.85**  | 4.45       | 4.65        | 4.60      |
> | Depth               | **4.80**  | 4.45       | 4.45        | 4.55      |
> | Foucs               | **4.95**  | 4.80       | 4.85        | 4.85      |
> | Coherence           | **4.75**  | 4.15       | 4.35        | 4.30      |
> | Fluency             | **4.80**  | 4.60       | 4.70        | 4.45      |
> | Average             | **4.83**  | 4.49       | 4.60        | 4.55      |
> | **Richness**        |
> | Avg. Fig. Num.      | 0.20      | -          | -           | 3.00      |
> | Avg. Table Num.     | 0.10      | -          | -           | 2.90      |
> | Total Avg.          | 0.37      | -          | -           | 2.81      |
>
> **Quiz-based Evaluation**
>
> |             | Concept   | Classification | History   | Alogrithm | Application | Profiling | Prediction | Topic Specific Score |
> | ----------- | --------- | -------------- | --------- | --------- | ----------- | --------- | ---------- | -------------------- |
> | AutoSurvey  | 0.538     | 0.133          | 1.000     | 0.222     | 0.600       | 0.318     | 0.552      | 2.36                 |
> | LLM×MR-V2   | **0.842** | 0.278          | **0.824** | **0.556** | **0.714**   | 0.526     | 0.710      | 2.22                 |
> | OpenAI-DR   | 0.744     | 0.474          | 0.647     | 0.333     | 0.636       | **0.562** | **0.757**  | **2.93**             |
> | SurveyForge | 0.444     | **0.500**      | 0.500     | 0.333     | 0.550       | 0.611     | 0.615      | 2.16                 |
>
> **Summary of Findings**
>
> From the results above, we observe that:
>
> 1. **SurveyBench generalizes well to disciplines beyond computer science.**
>    The relative ranking of methods in content-based evaluation and quiz-based evaluation remains largely consistent with the results reported in the CS domain (e.g., OpenAI-DR continues to achieve the highest content quality but comparatively weaker outline quality), indicating that the framework is not biased toward CS-specific writing conventions.
> 2. **Because many non-CS scientific fields evolve more gradually than CS/AI, their absolute scores tend to be higher,** which aligns with our findings on topic recency reported in the main paper. Non-CS topics usually have more stable and well-established research structures, making them easier for LLMs to organize and summarize, and this trend holds consistently across both content-based and quiz-based evaluation.
>
> These observations demonstrate that the conclusions derived from our testing framework are not confined to CS topics and generalize well to a broad range of scientific fields.
>
> [1] Biamonte J, Wittek P, Pancotti N, et al. Quantum machine learning[J]. Nature, 2017, 549(7671): 195-202.
>
> [2] Wasserman L. Topological data analysis[J]. Annual review of statistics and its application, 2018, 5(2018): 501-532.
>
> [3] Liu Y, Zhao T, Ju W, et al. Materials discovery and design using machine learning[J]. Journal of Materiomics, 2017, 3(3): 159-177.

---

> ### Author Response · Authors · 2025-11-24
>
> > Q2. Impact of topic type
>
> Thanks for this insightful question. To examine whether topic type affects LLM performance in survey generation, we categorize our topics into six representative topic types and report the averaged outline and content quality scores across four survey-generation methods.
>
> **Outline Quality**
>
> | Topic Type             | Coverage | Relevance | Structure | Avg. |
> | ---------------------- | -------- | --------- | --------- | ---- |
> | Reinforcement Learning | 3.44     | 3.94      | 3.73      | 3.70 |
> | Efficiency             | 4.00     | 4.33      | 3.92      | 4.08 |
> | LLM Application        | 3.85     | 4.15      | 3.85      | 3.95 |
> | LLM Core & Alignment   | 4.00     | 4.20      | 3.95      | 4.05 |
> | Perception             | 3.96     | 4.25      | 3.77      | 3.99 |
> | Multimodal & GenAI     | 3.78     | 4.20      | 3.83      | 3.94 |
>
> **Content Quality**
>
> | Topic Type             | Coverage | Depth | Focus | Coherence | Fluency | Avg. |
> | ---------------------- | -------- | ----- | ----- | --------- | ------- | ---- |
> | Reinforcement Learning | 3.62     | 3.31  | 3.94  | 3.88      | 3.81    | 3.71 |
> | Efficiency             | 4.00     | 3.75  | 4.25  | 3.75      | 4.25    | 4.00 |
> | LLM Application        | 3.98     | 3.98  | 4.50  | 4.05      | 4.35    | 4.17 |
> | LLM Core & Alignment   | 4.03     | 4.20  | 4.60  | 4.15      | 4.32    | 4.26 |
> | Perception             | 4.25     | 3.88  | 4.69  | 4.12      | 4.31    | 4.25 |
> | Multimodal & GenAI     | 4.20     | 4.10  | 4.65  | 4.08      | 4.30    | 4.26 |
>
> As shown in the table above, we observed two key findings:
>
> (1) LLM survey quality is relatively stable across different topic types. The scores are exhibit only small variations across categories, suggesting that topic type has only a minor impact on LLMs’ ability to produce high-quality survey drafts.
>
> (2) The RL-related topics and their corresponding human-written survey references selected in SurveyBench are notably recent (2025.09). LLMs perform worse on these topics due to the limited amount of mature literature available, providing further evidence that topic recency, rather than topic type, is the dominant factor influencing survey-writing performance.

---

> ### Author Response · Authors · 2025-11-26
>
> **Dear Reviewer 7uCU,**
>
> Thank you again for your thoughtful comment. As another reviewer raised questions regarding how our work differs from prior work, we would like to offer the following clarification to ensure that all reviewers share a clear and consistent understanding of these distinctions. The table below highlights where SurveyBench introduces capabilities that existing benchmarks do not support:
>
> | **Bench**              | **Topic Preparation**            | **Outline Quality** | **Content Quality** | **Reference Quality** | **Non-text Metric** | **Multi-setting Evaluation** | **Quiz-based Evaluation** |
> | ---------------------- | -------------------------------- | ------------------- | ------------------- | --------------------- | ------------------- | ---------------------------- | ------------------------- |
> | AutoSurvey[1]          | ×                                | ×                   | √ (3 sub-dims)      | √                     | ×                   | ×                            | ×                         |
> | SurveyForge[2]         | ×                                | √ (1 dim)           | √ (3 sub-dims)      | √                     | ×                   | ×                            | ×                         |
> | LLM×MR-V2[3]           | ×                                | ×                   | √ (4 sub-dims)      | √                     | ×                   | ×                            | ×                         |
> | **SurveyBench (Ours)** | **√ (Embedding-based curation)** | **√ (3 sub-dims)**  | **√ (5 sub-dims)**  | ×                     | **√ (Richness)**    | **√ (Dual-setting)**         | **√ (General+Specific)**  |
>
> As shown, SurveyBench differs from existing works in several key aspects:
>
> - **A rigorous topic-preparation pipeline**
> - **More fine-grained scoring dimensions (3 outline + 5 content vs. ≤4 in prior work)**
> - **Dual evaluation settings (with / without human reference)**
> - **First benchmark to evaluate visual richness (figures/tables)**
> - **First to introduce quiz-based, answerability-centered evaluation**
>
> We hope this clarifies how SurveyBench advances beyond prior work, and we welcome any further questions or feedback.
>
> ---
>
> [1] Wang Y, Guo Q, Yao W, et al. Autosurvey: Large language models can automatically write surveys[J]. Advances in neural information processing systems, 2024, 37: 115119-115145.
>
> [2] Yan X, Feng S, Yuan J, et al. Surveyforge: On the outline heuristics, memory-driven generation, and multi-dimensional evaluation for automated survey writing[C]//Proceedings of the 63rd Annual Meeting of the Association for Computational Linguistics (Volume 1: Long Papers). 2025: 12444-12465.
>
> [3] Wang H, Fu Y, Zhang Z, et al. LLM $\times$ MapReduce-V2: Entropy-Driven Convolutional Test-Time Scaling for Generating Long-Form Articles from Extremely Long Resources[J]. arXiv preprint arXiv:2504.05732, 2025.

---

### Official Review · Reviewer_epSS · 2025-11-01

**Soundness:** 3
**Presentation:** 3
**Contribution:** 2
**Rating:** 4
**Confidence:** 3

**Summary:**

This paper proposes a fine-grained evaluation framework named SurveyBench, designed to systematically assess the capability of large language models (and their agents) in automatically generating academic survey papers. The article points out that there is a significant quality gap between existing automatically generated surveys and those written by humans, and there is a lack of a rigorous, reader-needs-aligned benchmark to reveal these shortcomings.

**Strengths:**

1. The research background of this paper is the current lack of effective and rigorous benchmarks for evaluating the performance of LLMs in automatically generating academic surveys, which indeed represents a key area of focus in current research.
2. The design of SurveyBench is structured and systematic. It features a dual evaluation mode—"content-based" and "quiz-based." The latter ingeniously tests the depth and informational effectiveness of surveys by simulating readers' genuine knowledge-seeking needs. Meanwhile, its multi-dimensional evaluation framework (covering structure, content, and richness) comprehensively encompasses the key aspects of high-quality surveys.
3. The article utilizes SurveyBench to conduct experiments on multiple baselines, revealing the performance gap between automatically generated surveys and human-written ones. Additionally, it includes case studies and provides error analysis.

**Weaknesses:**

1. I think the most significant issue with this paper is the lack of innovation. Although SurveyBench incorporates different evaluation logics, each individual evaluation component lacks novelty. For instance, the "quiz-based evaluation" fundamentally relies on "LLM-as-a-Judge" and RAG (Retrieval-Augmented Generation) technologies, which are already being extensively explored and applied in the evaluation of long-text generation tasks. Similarly, the design of its evaluation metrics—such as coverage, depth, and coherence—follows conventional approaches commonly seen in this research field.
2. The paper lacks sufficient citations to substantiate its claims. With only 11 references cited in total—mostly documenting the origins of models, metrics, and survey frameworks—the arguments presented in the introduction remain inadequately supported and thus less convincing. For instance, the authors assert that "(3) existing LLM-as-judge evaluation struggles to capture the reader’s perspective or to probe whether a survey genuinely informs (e.g., technical depth) and inspires (e.g., forward-looking insights)," yet fail to provide references to justify this criticism.
3. The topics illustrated in the article appear to be exclusively from the field of computer science, which limits the generalizability of SurveyBench and the resulting conclusions. There is a lack of case studies analyzing surveys from other specialized domains. It remains unknown how well LLMs perform in generating literature reviews for disciplines with different document structures, writing conventions, and evaluation criteria, as well as how effective SurveyBench would be in such contexts.
4. The guidance for future work is somewhat underdeveloped. Although the article identifies shortcomings of current models (e.g., lack of detail, weak reasoning), it offers limited insight into how to address these challenges. The conclusion primarily summarizes findings but falls short of outlining a forward-looking research roadmap. Key questions remain unexplored: Should future efforts focus on improving retrieval quality, enhancing the reasoning modules of models, or designing novel generative architectures?

**Questions:**

See weakness

---

> ### Author Response · Authors · 2025-11-24
>
> **Dear Reviewer epSS,**
>
> Thank you for the thoughtful and detailed review. We truly appreciate your recognition of SurveyBench as well as your constructive suggestions, such as insufficient citations and limited topic fields. We hope our responses address your concerns and welcome any further questions or feedback.
>
> ---
>
> > W1. Lack of innovation
>
> Thanks for your insightful comment. We would like to clarify our innovation in the following aspects:
>
> **(1) Quiz-based evaluation**
>
> - General Quiz template: We derived them from a systematic analysis of high-quality surveys across multiple domains. We selected a set of well-cited and widely recognized surveys, examined their narrative structures and organizational patterns, and distilled the common high-information-density components. These distilled elements form the core of our quiz design, as they represent the defining characteristics that make these surveys “high-quality” in the first place.
> - The Topic-specific Quiz template evaluates the content gap between LLM-generated and high-quality human surveys using a hierarchical quality assurance framework. Paragraphs are filtered via a seven-dimensional assessment (length, completeness, formula/media density, substantive content, lexical diversity, and list avoidance) with progressive thresholds. Question–answer generation proceeds in three stages: standard generation with gradually relaxed thresholds, resampling of successful paragraphs to add questions while avoiding duplicates, and emergency minimum-standard generation if completion falls below 80%. All Q&A pairs must meet length requirements, be context-independent, provide substantive information, avoid speculation, and retain 10–20% keyword overlap with the source text. Deduplication uses Jaccard similarity (threshold 0.7), and robustness is ensured through paragraph-pool refresh.
>
> **(2) Content-based evaluation**
>
> - More fine-grained metrics: We design evaluation criteria that directly reflect the requirements of academic survey writing, emphasizing fine-grained assessment by decomposing outline quality into 3 sub-dimensions, and content quality into 5 sub-dimensions, allowing deeper diagnostic power during evaluation.
> - Richness: Non-text elements are central to high-quality surveys, but they are absent from all previous evaluation frameworks for survey generation. We introduce a new richness metric to quantify the presence of non-text elements like figures and tables.
> - Comprehensive evaluation setting: We propose a dual-mode evaluation protocol:
>
>   - Without human reference: suitable for open-domain topics;
>   - With human reference: aligns the model’s judgment to expert-written standards and avoids the widely reported score inflation common in LLM-as-a-Judge settings (evidenced by the contrast between Tables 7–8 and Table 3).
>
> **(3) New insights identified**
>
> We prepare a high-quality testing dataset using:
>
> - multi-field embedding clustering (title + abstract + topic embeddings)
> - pruning based on topic maturity, diversity, and diagnostic utility
> - sampling from 11,343 recent arXiv papers and 4,947 curated surveys
>
> This results in a batch of representative and diverse topics, forming the first benchmark targeted specifically at academic survey writing.
>
> With this high-quality dataset and our comprehensive evaluation framework, SurveyBench reveals several previously unreported findings, such as:
>
> - LLMs’ systematic weaknesses in algorithmic principles, performance insights, and associative reasoning;
> - The lack of synthesis and abstraction, where LLMs mainly rewrite retrieved content rather than integrating or prioritizing it;
> - Strong dependence on topic recency, with significantly lower performance on emerging domains.
>
> These insights have not been discussed in prior work and are made possible only through SurveyBench’s uniquely fine-grained and quiz-based evaluation design.

---

> ### Author Response · Authors · 2025-11-24
>
> > W2. Insufficient citations
>
> Thanks for pointing out this important issue regarding insufficient citations.
>
> We have carefully revised the manuscript and increased the total number of references to 20, adding several works that directly address the reviewer’s concern. The updated citations mainly include: (1) Studies on the limitations of LLM-as-judge, such as [1, 2]; (2) Recent benchmarks demonstrating that existing long-form evaluation fails to capture depth, synthesis, and conceptual reasoning, such as [3, 4].
>
> Now, the introduction is now more substantively supported and better grounded in existing literature. All critical statements are justified with citations rather than conceptual claims.
>
> We appreciate the reviewer’s suggestion, which significantly improved the scholarly rigor of our paper.
>
> [1] Li D, Jiang B, Huang L, et al. From generation to judgment: Opportunities and challenges of llm-as-a-judge[C]//Proceedings of the 2025 Conference on Empirical Methods in Natural Language Processing. 2025: 2757-2791.
>
> [2] Ye J, Wang Y, Huang Y, et al. Justice or prejudice? quantifying biases in llm-as-a-judge[J]. arXiv preprint arXiv:2410.02736, 2024.
>
> [3] Liu X, Dong P, Hu X, et al. Longgenbench: Long-context generation benchmark[J]. arXiv preprint arXiv:2410.04199, 2024.
>
> [4] Shaham U, Ivgi M, Efrat A, et al. ZeroSCROLLS: A zero-shot benchmark for long text understanding[J]. arXiv preprint arXiv:2305.14196, 2023.

---

> ### Author Response · Authors · 2025-11-24
>
> > W3. Limited topic fields
>
> Thanks for your thoughtful comment.
>
> To address this concern, we additionally expand SurveyBench with 10 topics and their corresponding human survey (such as [1, 2, 3]) from non-CS disciplines, including representative areas from Chemistry, Materials Science, Mathematics, Physics, Biology, and Medicine. These topics were selected to cover different writing conventions, structural norms, and reasoning styles commonly seen in natural sciences, thereby testing whether SurveyBench generalize beyond computer science.
>
> Below we report the extended evaluation results.
>
> **Content-based Evaluation**
>
> | Dimenson            | OpenAI-DR | AutoSurvey | SurveyForge | LLM×MR-V2 |
> | ------------------- | --------- | ---------- | ----------- | --------- |
> | **Outline Quality** |
> | Coverage            | 4.18      | 4.27       | 4.60        | **4.67**  |
> | Relevance           | 4.45      | 4.42       | 4.67        | **4.68**  |
> | Structure           | 4.05      | 3.98       | 4.40        | **4.45**  |
> | Average             | 4.23      | 4.22       | 4.56        | **4.60**  |
> | **Content Quality** |
> | Coverage            | **4.85**  | 4.45       | 4.65        | 4.60      |
> | Depth               | **4.80**  | 4.45       | 4.45        | 4.55      |
> | Foucs               | **4.95**  | 4.80       | 4.85        | 4.85      |
> | Coherence           | **4.75**  | 4.15       | 4.35        | 4.30      |
> | Fluency             | **4.80**  | 4.60       | 4.70        | 4.45      |
> | Average             | **4.83**  | 4.49       | 4.60        | 4.55      |
> | **Richness**        |
> | Avg. Fig. Num.      | 0.20      | -          | -           | 3.00      |
> | Avg. Table Num.     | 0.10      | -          | -           | 2.90      |
> | Total Avg.          | 0.37      | -          | -           | 2.81      |
>
> **Quiz-based Evaluation**
>
> |             | Concept   | Classification | History   | Alogrithm | Application | Profiling | Prediction | Topic Specific Score |
> | ----------- | --------- | -------------- | --------- | --------- | ----------- | --------- | ---------- | -------------------- |
> | AutoSurvey  | 0.538     | 0.133          | 1.000     | 0.222     | 0.600       | 0.318     | 0.552      | 2.36                 |
> | LLM×MR-V2   | **0.842** | 0.278          | **0.824** | **0.556** | **0.714**   | 0.526     | 0.710      | 2.22                 |
> | OpenAI-DR   | 0.744     | 0.474          | 0.647     | 0.333     | 0.636       | **0.562** | **0.757**  | **2.93**             |
> | SurveyForge | 0.444     | **0.500**      | 0.500     | 0.333     | 0.550       | 0.611     | 0.615      | 2.16                 |
>
> **Summary of Findings**
>
> From the results above, we observe that:
>
> 1. **SurveyBench generalizes well to disciplines beyond computer science.**
>    The relative ranking of methods in content-based evaluation and quiz-based evaluation remains largely consistent with the results reported in the CS domain (e.g., OpenAI-DR continues to achieve the highest content quality but comparatively weaker outline quality), indicating that the framework is not biased toward CS-specific writing conventions.
> 2. **Because many non-CS scientific fields evolve more gradually than CS/AI, their absolute scores tend to be higher,** which aligns with our findings on topic recency reported in the main paper. Non-CS topics usually have more stable and well-established research structures, making them easier for LLMs to organize and summarize, and this trend holds consistently across both content-based and quiz-based evaluation.
> 3. These findings collectively demonstrate that the conclusions drawn in our paper do not rely on CS topics alone, and SurveyBench serves as a domain-agnostic and extensible benchmark framework.
>
> [1] Biamonte J, Wittek P, Pancotti N, et al. Quantum machine learning[J]. Nature, 2017, 549(7671): 195-202.
>
> [2] Wasserman L. Topological data analysis[J]. Annual review of statistics and its application, 2018, 5(2018): 501-532.
>
> [3] Liu Y, Zhao T, Ju W, et al. Materials discovery and design using machine learning[J]. Journal of Materiomics, 2017, 3(3): 159-177.

---

> ### Author Response · Authors · 2025-11-24
>
> > W4. Lack of guidance for future work
>
> Thanks for this valuable suggestion.
>
> Articulating clear insights and challenges is crucial for guiding future progress in automatic survey generation. Our quiz-based evaluation reveals several key deficiencies in current LLM-generated surveys (section 4.1), and we outline corresponding research directions that could meaningfully advance the field:
>
> - Finding 1. Insufficient detail. Incorporating multi-hop reasoning modules or planning-based generation may further help models integrate detailed evidence during drafting.
> - Finding 2. Lack of associative reasoning. Building and Leveraging graph-based knowledge integration (e.g., taxonomy graphs) to assist models in forming deeper cross-concept analogies.
> - Finding 3. Deficient synthesis and abstraction. This limitation suggests the need for dedicated synthesis modules that go beyond extraction-style generation.

---

> ### Author Response · Authors · 2025-11-26
>
> **Dear Reviewer epSS,**
>
> Thank you again for your thoughtful comment. As another reviewer raised questions regarding how our work differs from prior work, we would like to offer the following clarification to ensure that all reviewers share a clear and consistent understanding of these distinctions. The table below highlights where SurveyBench introduces capabilities that existing benchmarks do not support:
>
> | **Bench**              | **Topic Preparation**            | **Outline Quality** | **Content Quality** | **Reference Quality** | **Non-text Metric** | **Multi-setting Evaluation** | **Quiz-based Evaluation** |
> | ---------------------- | -------------------------------- | ------------------- | ------------------- | --------------------- | ------------------- | ---------------------------- | ------------------------- |
> | AutoSurvey[1]          | ×                                | ×                   | √ (3 sub-dims)      | √                     | ×                   | ×                            | ×                         |
> | SurveyForge[2]         | ×                                | √ (1 dim)           | √ (3 sub-dims)      | √                     | ×                   | ×                            | ×                         |
> | LLM×MR-V2[3]           | ×                                | ×                   | √ (4 sub-dims)      | √                     | ×                   | ×                            | ×                         |
> | **SurveyBench (Ours)** | **√ (Embedding-based curation)** | **√ (3 sub-dims)**  | **√ (5 sub-dims)**  | ×                     | **√ (Richness)**    | **√ (Dual-setting)**         | **√ (General+Specific)**  |
>
> As shown, SurveyBench differs from existing works in several key aspects:
>
> - **A rigorous topic-preparation pipeline**
> - **More fine-grained scoring dimensions (3 outline + 5 content vs. ≤4 in prior work)**
> - **Dual evaluation settings (with / without human reference)**
> - **First benchmark to evaluate visual richness (figures/tables)**
> - **First to introduce quiz-based, answerability-centered evaluation**
>
> We hope this clarifies how SurveyBench advances beyond prior work, and we welcome any further questions or feedback.
>
> ---
>
> [1] Wang Y, Guo Q, Yao W, et al. Autosurvey: Large language models can automatically write surveys[J]. Advances in neural information processing systems, 2024, 37: 115119-115145.
>
> [2] Yan X, Feng S, Yuan J, et al. Surveyforge: On the outline heuristics, memory-driven generation, and multi-dimensional evaluation for automated survey writing[C]//Proceedings of the 63rd Annual Meeting of the Association for Computational Linguistics (Volume 1: Long Papers). 2025: 12444-12465.
>
> [3] Wang H, Fu Y, Zhang Z, et al. LLM $\times$ MapReduce-V2: Entropy-Driven Convolutional Test-Time Scaling for Generating Long-Form Articles from Extremely Long Resources[J]. arXiv preprint arXiv:2504.05732, 2025.

---

### Author Response · Authors · 2025-11-29
**General Response to all Reviewers**

**Dear Reviewers and Area Chair,**

We sincerely thank all reviewers for the constructive and insightful feedback. Your comments greatly helped us improve the clarity, rigor, and scope of our work. Below we summarize the major concerns raised by the reviewers, grouped by theme for clarity and completeness.

1. **Limited Topic Fields:** We expand SurveyBench with 10 additional non-CS topics (Chemistry, Materials, Math, Physics, Bio, Medicine) to test domain transferability. Results show similar method ranking and trends, demonstrating SurveyBench generalizes beyond CS.
2. **LLM-as-Judge Reliability:** We evaluate robustness using APAD across three runs with 95% bootstrapped CIs. Both content- and quiz-based evaluation remain stable, supporting SurveyBench’s reliability as an LLM-as-judge framework.
3. **Impact of Topic Type:** We categorize our topics into six representative topic types and find score variations are minor across categories.
4. **Model Leakage:** Most methods are restricted to abstracts and titles only, preventing exposure to survey texts. Citation analysis further shows very low survey-reference ratios, indicating leakage is minimal.
5. **Insufficient Citations:** We expand the reference list to 20+ works, adding missing literature on LLM-as-judge limitations and long-form evaluation challenges, ensuring claims are well-supported and academically grounded.
6. **Guidance for Future Work:** Based on quiz-based insights, we summarize three forward directions: improving detail via multi-hop planning, enhancing associative reasoning via graph-based knowledge, and enabling deeper synthesis beyond extraction-style generation.
7. **Crowded Tables and Figures:** We will refine visual layout (e.g., enlarging fonts, clarifying legends for Fig. 5) to further improve presentation clarity.

Besides, as reviewer Mxrn raised questions regarding how our work differs from prior work, we would like to offer the following clarification to ensure that all reviewers share a clear and consistent understanding of these distinctions. The table below highlights where SurveyBench introduces capabilities that existing benchmarks do not support:

| **Bench**              | **Topic Preparation**            | **Outline Quality** | **Content Quality** | **Reference Quality** | **Non-text Metric** | **Multi-setting Evaluation** | **Quiz-based Evaluation** |
| ---------------------- | -------------------------------- | ------------------- | ------------------- | --------------------- | ------------------- | ---------------------------- | ------------------------- |
| AutoSurvey          | ×                                | ×                   | √ (3 sub-dims)      | √                     | ×                   | ×                            | ×                         |
| SurveyForge         | ×                                | √ (1 dim)           | √ (3 sub-dims)      | √                     | ×                   | ×                            | ×                         |
| LLM×MR-V2           | ×                                | ×                   | √ (4 sub-dims)      | √                     | ×                   | ×                            | ×                         |
| **SurveyBench (Ours)** | **√ (Embedding-based curation)** | **√ (3 sub-dims)**  | **√ (5 sub-dims)**  | ×                     | **√ (Richness)**    | **√ (Dual-setting)**         | **√ (General+Specific)**  |

As shown, SurveyBench differs from existing works in several key aspects:

- **A rigorous topic-preparation pipeline**
- **More fine-grained scoring dimensions (3 outline + 5 content vs. ≤4 in prior work)**
- **Dual evaluation settings (with / without human reference)**
- **First benchmark to evaluate visual richness (figures/tables)**
- **First to introduce quiz-based, answerability-centered evaluation**

We again thank all reviewers for the valuable comments and thoughtful assessment. We believe these discussions help establish a clearer understanding of our contributions, and we hope our responses address all concerns satisfactorily.

---

### Meta-Review · Area_Chair_qDsV · 2025-12-26

**Summary:**

The paper introduces SurveyBench, a fine-grained, quiz-driven evaluation framework designed to rigorously assess the quality of LLM-generated academic surveys. Built from thousands of recent arXiv papers and high-quality human surveys, it evaluates outline structure, content synthesis, and non-textual richness, according to a dual-mode evaluation protocol. Experiments show that current LLM4Survey approaches perform significantly worse than humans, highlighting substantial gaps in survey quality.

The strengths include 1) the benchmark construction is needed for this research area; 2) the design of SurveyBench is structured and systematic; 3) the experiments reveals some interesting findings; 4) the evaluation designs are reasonable. However, the reviewer concerns include:
1. The novelty is limited and the benchmark has significant overlap with existing works, e.g. SurveyForge (epSS, Mxrn, gxsR);
2. Missing sufficient references (epSS);
3. Missing sufficient topic coverage beyond compute science domain (epSS, 7uCU);
4. The guidance for future work is somewhat underdeveloped (epSS);
5. The heavy reliance on LLM-as-judge exhibits limitations in discrimination and stability (7uCU, gxsR);
6. The richness metric invites superficial additions instead of evidence-grounded visuals (gxsR);
7. Model leakage: Instructing agents not to consult existing surveys is hard to enforce, and many human surveys used as references are likely in pretraining corpora (gxsR).

According to the rebuttal, the AC thinks most of the concerns are only partially addressed. For example:
- #1: This is a big concern for three reviewers. Although the rebuttal provided detailed response, e.g. claiming the proposed benchmark is more diverse, more finegrained, more metrics, etc., the AC thinks this paper doesn't have enough differences from previous works, e.g. SurveyForge, but more like an extension work.
- #2: The number of 11 references is unreasonable low for a modern DL/LLM paper, especially it is not on a fundamental new topic. Although the rebuttal mentioned to add 4 more references, the number is still very low. Sufficient literature review is required!
- #3: Although the rebuttal claimed to add more topics, many details on the added topics are missing, e.g. where are the statistics, how were they collected/curated? This is a big change to the paper, but the provided details are not enough to evaluate the reliability of the new results in such short rebuttal time.
- #5: Although the inter-model consistency was evaluated, what are the models evaluated?
- #6: The issue doesn't seem to be addressed if the model generates unnecessarily more visuals.
- #7: The rebuttal doesn't address what if the human surveys are being used during training stages of those LLMs.

Given these, the AC doesn't think this paper is ready to be published by ICLR yet. The authors are strongly suggested to revise the paper following the reviewers' feedback.

**Reviewer Concerns:**

However, the reviewer concerns include:
1. The novelty is limited and the benchmark has significant overlap with existing works, e.g. SurveyForge (epSS, Mxrn, gxsR);
2. Missing sufficient references (epSS);
3. Missing sufficient topic coverage beyond compute science domain (epSS, 7uCU);
4. The guidance for future work is somewhat underdeveloped (epSS);
5. The heavy reliance on LLM-as-judge exhibits limitations in discrimination and stability (7uCU, gxsR);
6. The richness metric invites superficial additions instead of evidence-grounded visuals (gxsR);
7. Model leakage: Instructing agents not to consult existing surveys is hard to enforce, and many human surveys used as references are likely in pretraining corpora (gxsR).

Many of them are not well addressed, including #1, #2, #3, #5, #6, and #7.

**Reviewer Scores:**

The original scores are 4 (epSS), 4 (7uCU), 2 (Mxrn), 6 (gxsR). Given most the concerns were not well addressed, the AC doesn't think the reviewers will change their scores.

---

### Decision · Program_Chairs · 2026-01-26

Reject